# Large off-diagonal magnetoelectricity in a triangular $Co^{2+}$-based collinear antiferromagnet

Xianghan Xu [1] ✉, Yiqing Hao [2], Shiyu Peng[3], Qiang Zhang [2], Danrui Ni [1], Chen Yang[1], Xi Dai [3], Huibo Cao [2] & R. J. Cava[1] ✉

Magnetic toroidicity is an uncommon type of magnetic structure in solid-state materials. Here, we experimentally demonstrate that collinear spins in a material with $R\text{-}3$ lattice symmetry can host a significant magnetic toroidicity, even parallel to the ordered spins. Taking advantage of a single crystal sample of $CoTe_6O_{13}$ with an $R\text{-}3$ space group and a $Co^{2+}$ triangular sublattice, temperature-dependent magnetic, thermodynamic, and neutron diffraction results reveal A-type antiferromagnetic order below 19.5 K, with magnetic point group $-3'$ and $\mathbf{k} = (0,0,0)$. Our symmetry analysis suggests that the missing mirror symmetry in the lattice could lead to the local spin canting for a toroidal moment along the $c$ axis. Experimentally, we observe a large off-diagonal magnetoelectric coefficient of 41.2 ps/m that evidences the magnetic toroidicity. In addition, the paramagnetic state exhibits a large effective moment per $Co^{2+}$, indicating that the magnetic moment in $CoTe_6O_{13}$ has a significant orbital contribution. $CoTe_6O_{13}$ embodies an excellent opportunity for the study of next-generation functional magnetoelectric materials.

Two of the fundamental properties in solids, magnetism and electricity, generally have distinct microscopic origins[1,2]. Nevertheless, their cross-coupling, i.e., electric field control of magnetization or magnetic field control of electric polarization, can be realized in magnetoelectric materials[3–8]. Magnetoelectric materials have attracted significant recent research attention because they are ideal venues for exploring the interplay among charge, lattice, orbital, and spin degrees of freedom in the quantum regime, as well as an extensive prospect for applications in next-generation technologies such as logic circuits, sensors, actuators, memory devices, and energy-storage systems[9–15].

One strategy to achieve a strong coupling of magnetism and electricity is to fabricate heterogeneous structures consisting of a ferroelectric material and a ferromagnetic material. However, the multi-phase aspects of such systems, the materials interfaces, the substrates, and their limited sizes often bring nuisances into the experimental characterizations and theoretical understanding of such systems. On the other hand, symmetry analysis suggests that certain combinations of lattice structure and magnetic structure allow coupling between them even in a single-phase material. Using $Cr_2O_3$ (the first magnetoelectric material discovered) as an example[16], while it has a centrosymmetric lattice, adding antiferromagnetic spins order breaks the inversion center, and makes spatial inversion plus time reversal a new invariant symmetry operation, locking together the flipping of magnetization and induced polarization, i.e., a linear magnetoelectric effect. Furthermore, theoretical study suggests that, in materials with in-plane spins forming a head-to-tail vortex-like magnetic structure (magnetic toroidicity), applying an in-plane magnetic field could induce an in-plane electric polarization perpendicular to that magnetic field, i.e., an off-diagonal magnetoelectricity[17,18]. This exotic behavior, if present, adds both theoretical and application value to materials in the magnetoelectric family. For example, a magnetic vortex-like structure possessing toroidal moments, and off-diagonal magnetoelectricity has been experimentally demonstrated in $BaCoSiO_4$[19,20]. Magnetic toroidicity has also been experimentally

[1]Department of Chemistry, Princeton University, Princeton, NJ 08544, USA. [2]Neutron Scattering Division, Oak Ridge National Laboratory, Oak Ridge, TN 37831, USA. [3]Department of Physics, Hong Kong University of Science and Technology, Hong Kong, China. ✉e-mail: xx8060@princeton.edu; rcava@princeton.edu

observed in a few antiferromagnets with collinear spins, such as LiCoPO4[21,22], LiNiPO4[23], and CaMnGe2O6[24], for which the origin of magnetic toroidicity seems is not straightforward. Therefore, the experimental exploration of off-diagonal magnetoelectric materials having collinear spins and the illustration of the underlying physics picture that how collinear spins act to indue toroidicity is urgently needed for designing next-generation magnetoelectric materials with functional application potential.

Thus, here, we synthesize polycrystalline and single crystal samples of CoTe6O13, a material that contains a $Co^{2+}$-based triangular sublattice. Our powder and single crystal x-ray diffractions find an $R$-3 crystal structure for this material, consistent with previous reports[25] with all mirror symmetries broken, creating a ferro-rotation type of structural order. Our anisotropic magnetization and thermodynamic measurements on CoTe6O13 single crystals unveil the presence of an antiferromagnetic transition at a Néel temperature ($T_N$) around 20 K, a highly anisotropic $S = \frac{1}{2}$ ground state, and a large effective moment in the paramagnetic state, which is further supported by our first-principle calculation results. We perform neutron diffraction experiments and demonstrate that the magnetic ground state is an A-type collinear antiferromagnetic order with spins parallel/antiparallel to the $c$ axis ($\mathbf{k} = (0,0,0)$), and that the magnetic point group is -3'. We find a clear dielectric anomaly at $T_N$, indicating strong correlations between lattice structure and magnetism. Our pyroelectric and magnetoelectric current measurements unambiguously show an electric polarization developing below $T_N$. Remarkably, we observe both diagonal ($P//H$) and off-diagonal ($P \perp H$) linear magnetoelectricity in the entire −9 T to 9 T range, with a large off-diagonal linear magnetoelectric coefficient around 41.2 ps/m. Our symmetry analysis emphasizes the critical role that the structural ferro-rotation order plays in enabling the off-diagonal magnetoelectricity observed. Our results suggest that

CoTe6O13 is an excellent venue for the exploration of interplayed charge, lattice, orbital, and spin degrees of freedom in the quantum regime. Our analysis also establishes a symmetry connection between structural ferro-rotation and magnetic ferro-toroidal orderings in a single-phase material.

## Results
### Crystal structure

Figure 1a displays the Rietveld refinement result of the powder XRD pattern taken on crushed CoTe6O13 crystals. The rhombohedral lattice parameters are found to be $a = 10.1593(8)$ Å and $c = 18.9739(6)$ Å. A photograph of an as-grown CoTe6O13 crystal is shown in the inset. Single-crystal XRD was performed on a small piece of single crystal to confirm its identity, and the crystal structure obtained from refinement of the diffraction data, which is in agreement with a previous report[25], is displayed in Fig. 1b. The crystallographic data determined in this study for CoTe6O13, including atomic positions, site occupancies, and refined thermal parameters, are listed in Table SI and Table SII. The unit cell parameters obtained from single crystal diffraction, $a = 10.1660(13)$ Å and $c = 18.981(3)$ Å, match well with the ones obtained from the ground crystal powder XRD. The $Co^{2+}$ ions form triangular planar layers as illustrated by the top view Fig. 1c. The diffraction result indicates that the material has a space group $R$-3 (No. 148), again consistent with the previous report[25]. A ferro-rotation (also known as ferro-axial) order means a head-to-tail arrangement of electric dipoles (**D**), and the order parameter is defined by $\mathbf{r} \times \mathbf{D}$, i.e., an axial vector. The existence of a ferro-rotation order requires that all the mirror planes parallel to the axial vector are broken. In $R$-3, all the mirror symmetries are broken, and a ferro-rotation type structural order is allowed. Consequentially, the $Co^{2+}$ octahedra are distorted in this lattice. As shown in Fig. 1c, the top oxygen triangle and bottom

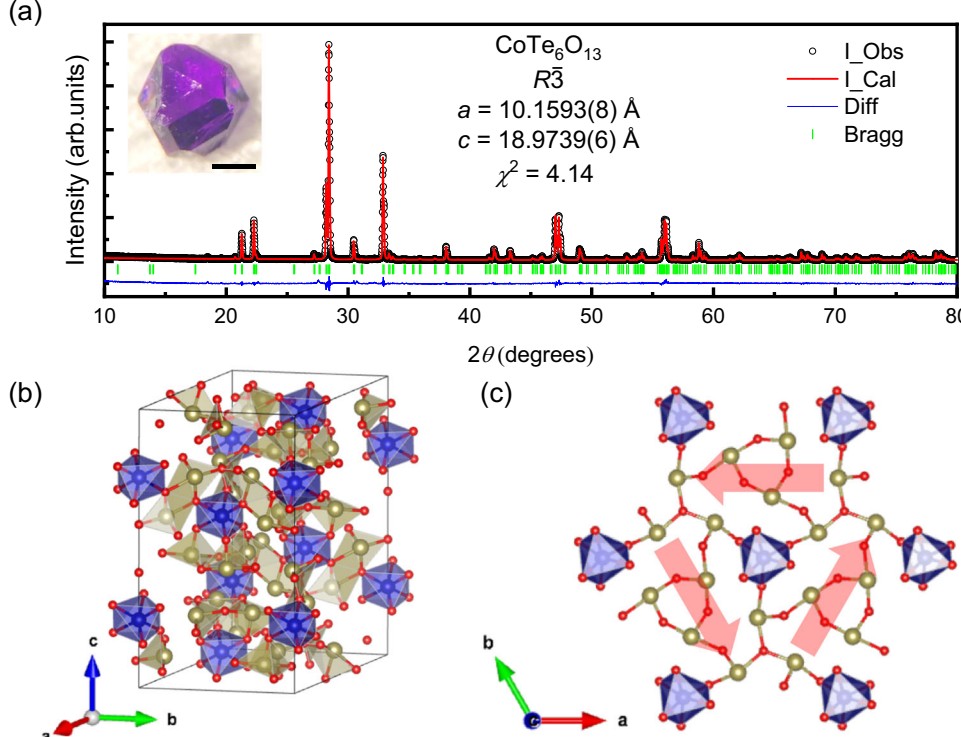

**Fig. 1 | Crystal structure and ferro-rotation. a** Rietveld refinement of the powder XRD pattern taken on crushed CoTe6O13 crystals (parameters in Table SIII). Open circles, red curve, blue curve, and green ticks display observed intensity (I_Obs), calculated intensity (I_Cal), intensity difference (Diff), and Bragg diffraction positions (Bragg). Inset displays a photo of a CoTe6O13 crystal. The black scale bar represents 500 μm. **b** Room temperature crystal structure of CoTe6O13. Co, Te, and O atoms are represented by blue, yellow, and red spheres, respectively. **c** Top view of a $Co^{2+}$ triangular layer. The red arrows display the local electric dipoles that form a ferro-rotation structure order.

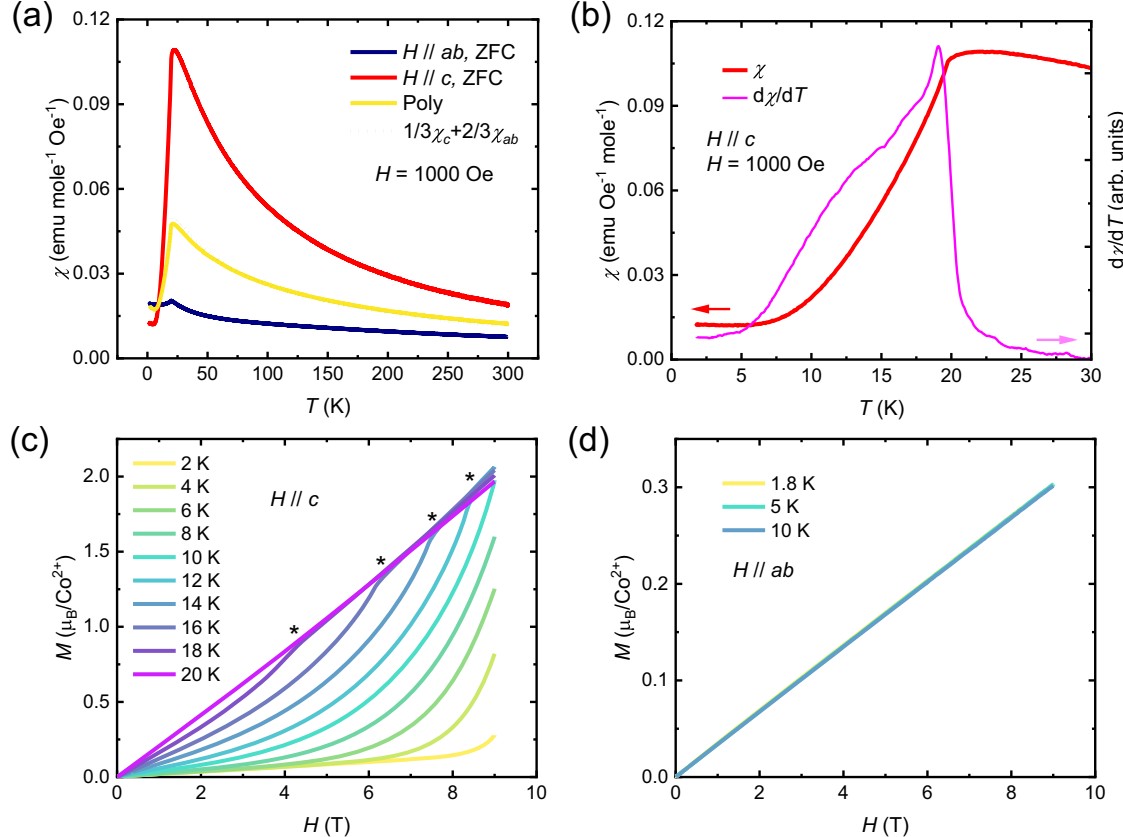

**Fig. 2 | Magnetization vs. temperature and field. a** Magnetic susceptibility ($\chi$) of $CoTe_6O_{13}$. Red, blue, yellow, and green curves display the magnetic susceptibility versus temperature ($T$) parallel to $c$ axis of a single crystal, perpendicular to $c$ axis of a single crystal, of a polycrystalline sample, and poly-averaged susceptibility of the single crystal data, respectively. **b** Low-temperature regime of the susceptibility along the $c$ axis. The pink curve displays the derivative of the susceptibility ($d\chi/dT$). **c** Magnetization ($M$) along the $c$ axis versus magnetic field ($H$) measured at various temperatures marked by different colors shown in the legend. Asterisks mark the meta-magnetic transitions. **d** Magnetization perpendicular to the $c$ axis versus field at 1.8 K, 5 K, and 10 K, displayed by the yellow, cyan, and light blue curves, respectively.

oxygen triangle are twisted along the $c$ axis in each $Co^{2+}$ octahedron, which breaks all the local mirror symmetries as well. This ferro-rotation type of distortion of an octahedron is also observed in $R$-3 ilmenites ($MTiO_3$ for M = Mn, Fe, Co, Ni)[26]. However, $CoTe_6O_{13}$ shows a much larger distortion from an ideal octahedron (Fig. S1). Consequentially, unlike $NiTiO_3$, which shows multiple ferro-rotation domains linked by mirror symmetry at room temperature[27–29], X-ray and neutron diffraction results consistently suggest a mono ferro-rotation domain in millimeter-size $CoTe_6O_{13}$ crystals, possibly to avoid what would be an extremely large lattice mismatch at domain walls.

## Magnetism and magnetic structure

In the $ab$ plane, the magnetic $Co^{2+}$ ions form a triangular lattice (Fig. 1c). As shown in Fig. 2a, anisotropic magnetic susceptibility ($\chi_c$ and $\chi_{ab}$) is observed. In the paramagnetic state, $\chi_c$ is significantly larger than $\chi_{ab}$, suggesting an Ising single-ion anisotropy. Upon cooling, a clear peak appears at around 20 K along both directions, suggesting antiferromagnetic long-range order. The averaged single crystal susceptibility by $\frac{1}{3}\chi_c + \frac{2}{3}\chi_{ab}$ (green dashed curve in Fig. 2a) overlaps with the susceptibility measured on the polycrystalline sample (yellow curve in Fig. 2a), which confirms the consistent sample quality. Figure 2b displays the derivative of susceptibility along the $c$ axis. It is clear that in addition to the transition at ~ 20 K, another broad feature exists from around 5 K to 15 K. A similar broad feature has been reported in the magnetic susceptibility of spinel $Co_3O_4$[30], whose origin is believed to be a short-range incommensurate order evidenced by the diffuse scattering in neutron experiments[31,32]. Isothermal magnetization

versus field ($M$-$H$) data with the magnetic field applied along the $c$ and $ab$ directions are shown in Fig. 2c, d, respectively. A meta-magnetic transition (marked by asterisks) appears when applying field along the $c$ axis, which is very likely a spin-flop transition. No hint of a transition is detected up to 9 T when the applied field is parallel to the $ab$ plane. This anisotropic behavior suggests that the ordered spins in the antiferromagnetic state primarily point along $c$. Figure 3 summarizes the inverse susceptibilities. The fitting of the high-temperature range (200 K to 300 K) paramagnetic susceptibilities using Curie-Weiss law $\chi = \frac{C}{T - T_{CW}}$, where $C = \frac{1}{8}\mu_{eff}^2$, gives $T_{CW\_c} = 16.2$ K, $\mu_{eff\_c} = 6.5$ $\mu_B/Co^{2+}$, for the $c$ direction, $T_{CW\_ab} = -187.5$ K, $\mu_{eff\_ab} = 5.4$ $\mu_B/Co^{2+}$, for the $ab$ direction, and $T_{CW\_poly} = -63.5$ K, $\mu_{eff\_poly} = 5.9$ $\mu_B/Co^2$, for the polycrystalline sample. The effective moment obtained from the $CW$ fitting is much larger than the spin-only moment 3.87 $\mu_B$ of $Co^{2+}$ ($S = 3/2$), and it is close to the theoretical value of magnetic moment with a full orbital contribution, 6.63 $\mu_B$. The large effective moment motivated us to test the material's composition independently of our crystal structure determination and the one reported previously. As shown in Fig. S2, Energy-dispersive x-ray spectroscopy (EDS) elemental analysis rules out the possibility that there is more $Co^{2+}$ in the $CoTe_6O_{13}$ sample than is observed in the structural analyses. As shown in Fig. S3, our X-ray photoelectron spectroscopy (XPS) result confirms the 2+ oxidation state of cobalt in $CoTe_6O_{13}$. Therefore, it can be concluded that the unusually large effective moment in the paramagnetic regime of $CoTe_6O_{13}$ is intrinsic, therefore indicating that there may be a significant orbital contribution to the large effective magnetic moment of $Co^{2+}$ ions in $CoTe_6O_{13}$.

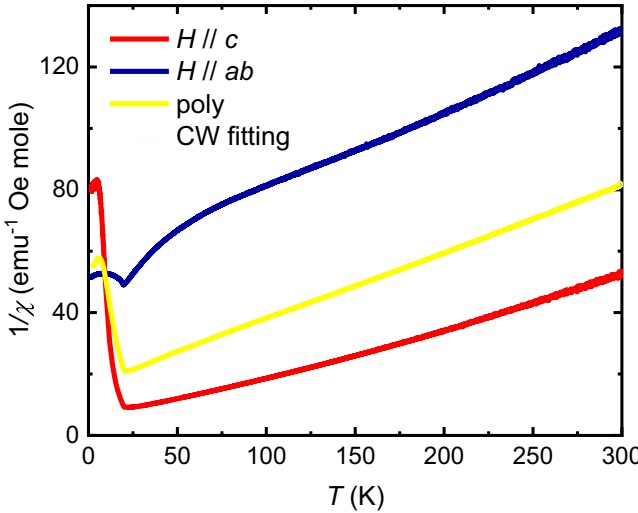

**Fig. 3 | The inverse susceptibilities.** Red, blue, yellow, and green curves display the inverse magnetic susceptibility ($1/\chi$) versus temperature parallel to $c$ axis of a single crystal, perpendicular to $c$ axis of a single crystal, and of a polycrystalline sample. The Curie-Weiss (CW) fitting is performed in the 200 K to 300 K range.

**Table 1 | Calculated magnetic moments and spin and orbital components**

| Principal axis | Total magnetic moment ($\mu_B$) | Orbital magnetic moment ($\mu_B$) | Spin magnetic moment ($\mu_B$) | Percentage of orbital magnetic moment |
|---|---|---|---|---|
| $x$ | 0.97 | 0.12 | 0.86 | 12% |
| $y$ | 0.97 | 0.12 | 0.86 | 12% |
| $z$ | 6.98 | 1.97 | 5.0 | 28% |

## Theoretical calculation

To understand the formation mechanism of local moments on the Co sites, an atomic model is derived with all the parameters like the strength of crystal field and spin orbital coupling being extracted from the first-principle calculation. The model reads,

$$\hat{H}_{atom} = \hat{H}_{kanamori} + \hat{H}_{cf} + \hat{H}_{soc} \tag{1}$$

where the first term includes both the Coulomb interaction and Hund's coupling, the second term signifies crystal field interaction (CFI) and the last term denotes spin-orbital coupling (SOC). Our first-principle calculations, conducted within the Vienna ab initio simulation package (VASP)[33,34], are based on the density functional theory (DFT) and utilize the Perdew-Burke-Ernzerhof (PBE) generalized gradient approximation (GGA) for the exchange-correlation potential[34]. For the plane wave basis, we have set an energy cutoff at 550 eV and adopted a fine $k$-point mesh of 7×7×7. Subsequently, we have interpolated a tight-binding model using the Wannier90 package, which allows us to extract values for the CFI ($\Delta_{Oct} \approx 0.9$ eV, $\Delta_{Tri} \approx 20$ meV) and SOC ($\lambda_{soc} \approx 60$ meV). Regarding the many-body interaction, we have opted for values of $U = 5$ eV and $J = 0.8$ eV.

By diagonalizing the atomic Hamiltonian, we derive all of the eigenstates that exhibit Kramers' degeneracy within the Fock subspace of seven electrons. Here we focus on the ground state subspace, often referred to as the pseudo-spin subspace, which is more than 20 meV below the first excited state. Within this pseudo-spin space, we establish three principal axes $\tilde{x}, \tilde{y}, \tilde{z}$, [refer to Supplement Sec 1]. Axes $\tilde{x}$ and $\tilde{y}$ lie on the ab plane and axis $\tilde{z}$ coincides with the $C3$ rotational axis of the conventional lattice. Using Curie-Weiss theorem [see Supplement Sec 1], we discern the effective magnetic moment for these principal axes. As can be observed in Table 1, the orbital component significantly contributes to the total magnetic moment, namely 1.97 $\mu_B$ (28%) for local moments along the $\tilde{z}$ axis and 0.12 $\mu_B$ (12%) for $\tilde{x}/\tilde{y}$ axis, which confirms the experimental magnetism observations in paramagnetic region. The substantial orbital contribution may result from two stages. Firstly, the Hunds' coupling outcompetes the octahedron splitting, allowing the $t_{2g}^5 e_g^2$ configurations to primarily occupy the lower energy eigenstates, thus enabling the emergence of a large orbital magnetic moment because the $t_{2g}$ orbitals are not fully occupied. Subsequently, the SOC dominates trigonal crystal field interaction within the $t_{2g}$ resulting in a stable complex orbital and large orbital

moment. Moreover, the effective magnetic moment exhibits considerable anisotropy, with a large value of 6.98 $\mu_B$ along $c$ axis and very small value of 0.97 $\mu_B$ within the $ab$ plane. The theoretical value of the $c$ axis moment is in good agreement with the experimental value from the $CW$ fitting, while the theoretical $ab$ plane moment is much smaller than the $CW$ fitting result. We believe that this deviation is caused by non-negligible magnetic interactions in the $CW$ fitting temperature range (200 K to 300 K), which is evidenced by the large absolute value of $CW$ temperature (−187.5 K) in the $ab$ plane.

## Heat capacity

To quantitatively study the thermodynamics of the antiferromagnetic transition and determine the low-temperature Co²⁺ spin degrees of freedom, heat capacity $C_p$ versus temperature measurements at various fields were performed on a polycrystalline CoTe₆O₁₃ sample. As shown in Fig. 4a, the heat capacity shows a clear peak at $T_N \sim 19.5$ K, with peak height close to the theoretical value for a 3D antiferromagnetic transition as predicted by the mean-field theory[35], and the non-magnetic contribution is obtained by measuring the heat capacity of an isostructural non-magnetic compound MgTe₆O₁₃. Applying magnetic field gradually suppresses the anomaly in the heat capacity. The magnetic contribution ($C_{mag}$) to the heat capacity is obtained by subtracting the heat capacity of MgTe₆O₁₃ from the total heat capacity of CoTe₆O₁₃, and Fig. 4b plots the resulting $C_{mag}/T$ as a function of temperature. Besides the peak at $T_N$, another broad feature at around 4 K is observed. The origin of this feature is currently unclear. The magnetic entropy at zero field is obtained by integrating $C_{mag}/T$ (Fig. 4c). It saturates above 30 K and reaches 5.75 J mole-Co⁻¹ K⁻¹. This value is close to the total magnetic entropy for an effective $S = ½$ spin, $Rln2$. Co²⁺ $3d^7$ is a Kramer ion, and the interplay of crystal electric field and spin-orbit coupling could result in an effective $S = ½$ ground state, which has been observed in compounds with octahedral Co²⁺ such as Na₂Co₂TeO₆[36], Na₃Co₂SbO₆[37], and K₂Co₂(SeO₃)₃[38].

## Neutron diffraction

To investigate the magnetic structure below $T_N$, temperature-dependent neutron diffraction experiments were performed on a CoTe₆O₁₃ single crystal. For crystallographic space group $R$-3 and propagation vector $\mathbf{k} = (0, 0, 0)$, there are four possible magnetic irreducible representations, i.e., (i) $mGM_1^+$, (ii) $mGM_2^+GM_3^+$, (iii) $mGM_1^-$, and (iv) $mGM_2^-GM_3^-$, resulting in the magnetic space groups $R$-3, $P$-1, $R$-3′, and $P$-1′, respectively. $R$-3 and $P$-1 belong to the ferromagnetic space group, which are not consistent with the observed antiferromagnetic ordering. Also, they do not break spatial inversion and do not allow linear magnetoelectricity, thus can be ruled out. Both $R$-3′ and $P$-1′ describe an antiferromagnetic ordering, while the spins in $R$-3′ are parallel/antiparallel to the three-fold axis, but $P$-1′ breaks the three-fold symmetry. Refining the magnetic structure using the $P$-1′ model obtained negligible in-plane magnetic moments (0.2(2) $\mu_B$), which suggests that the ordered moments are mostly along $c$. The absence of $\sigma_{13}$ in the experimental magnetoelectric tensor (Fig. S6) also provides evidence against the $P$-1′ model. Overall, $R$-3′ (*BNS #148.19*) is the appropriate model to describe the antiferromagnetic ordering in CoTe₆O₁₃.

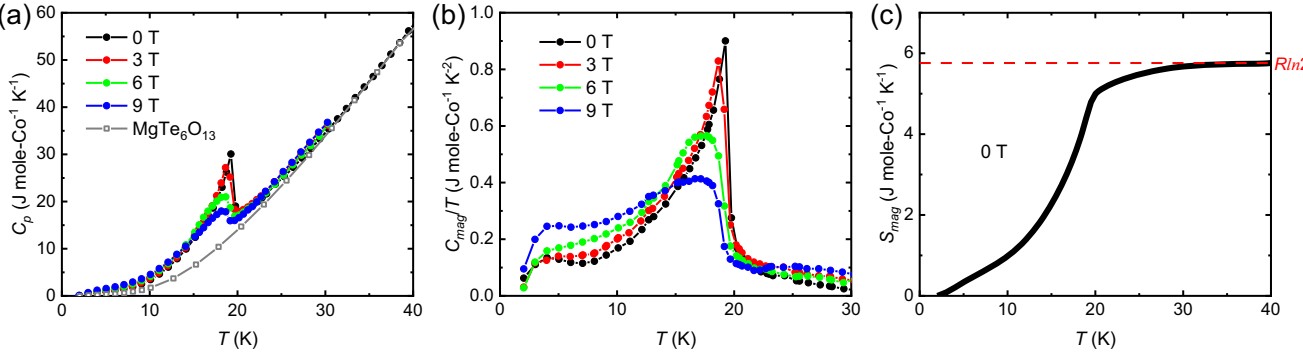

**Fig. 4 | Thermodynamics. a** The heat capacity ($C_p$) of CoTe$_6$O$_{13}$ from 1.8 K to 40 K measured at 0 T, 3 T, 6 T, and 9 T field, displayed by black, red, green, blue dots and curves, respectively. The gray open squares display the heat capacity of the non-magnetic reference compound MgTe$_6$O$_{13}$. **b** The magnetic heat capacity over temperature ($C_{mag}/T$) obtained from subtracting the non-magnetic contribution. **c** The magnetic entropy ($S_{mag}$) versus temperature at 0 T obtained by integrating $C_{mag}/T$. The horizontal red dashed line shows the value of $R\ln2$.

Of the 23 measured Bragg peaks, 9 were found of non-negligible magnetic contributions and are summarized in Table SV. Fig. 5a displays the magnetic structure solved by neutron diffraction. In each triangular Co$^{2+}$ layer (the *ab* plane), the spins are ferromagnetically aligned. Along the *c* axis, layers with alternatingly up and down spins are stacked, forming a so-called A-type antiferromagnetic structure. The measured ordered magnetic moment is 4.4(1) μ$_B$/Co$^{2+}$ at 5 K, which is much larger than the measured ordered moment of most reported Co$^{2+}$ compounds, compared, for example, to 1.8 μ$_B$ in Na$_3$Co$_2$SbO$_6$ at 1.5 K[37], 2.7 μ$_B$ in BaCoSiO$_4$ at 1.8 K[19], and 3.5 μ$_B$ in BiCoPO$_5$ at 1.5 K[39]. Therefore, the neutron diffraction study confirms the validity of the large effective moment obtained from the *CW* fitting of the paramagnetic regime. The (0, 1, −1) magnetic peak intensity as a function of temperature is shown in Fig. 5b, exhibiting an abrupt increase at $T_N$. The peak intensity is fitted using the formula $I–I_0 = A(1 − T/T_N)^{2\beta}$ with the critical exponent $\beta = 0.28(5)$. The $\beta$ value lies between 0.125 expected for 2D Ising moments and 0.325 expected for 3D Ising moments and is close to 0.25 expected for a mean-field tricritical model, which is understandable since both susceptibility (Fig. 2b) and pyroelectric current density (section vi.) show hints of an underlying phase transition below $T_N$. Notably, due to limited data points collected near $T_N$, we believe that our fitting only serves as a guidance for the measured data rather than yielding a definitive critical exponent $\beta$. As shown in Fig. 5c, d, the refinement of neutron diffraction data above $T_N$ and below $T_N$ both achieve great agreement between calculations and observations.

To ascertain the adopted magnetic model, neutron powder diffraction (NPD) experiments are performed on an around 5-gram polycrystalline sample of CoTe$_6$O$_{13}$ at three temperatures, i.e., 40 K, 14 K and 8 K. As shown in Fig. S4, additional diffraction intensities below $T_N$ demonstrate magnetic diffractions from the ordered magnetic state. Table SVI summarizes the Bragg indices of observed magnetic peaks, and all reflections can be indexed with the crystallographic unit cell, which is consistent with a wave vector **k** = (0, 0, 0). Using the *R*-3′ (BNS #148.17) model, the NPD patterns can be well fitted, and the refinement results are displayed in Fig. S5. The obtained ordered magnetic moment ($m_z$) of Co$^{2+}$ is 3.81(4) μ$_B$ at 14 K and 4.43(4) μ$_B$ at 8 K, which is quite close to the value obtained from the single-crystal neutron diffraction. Overall, the single-crystal and powder neutron diffraction results consistently manifest the **k** = (0, 0, 0) and -3′ magnetic ordering.

## Magnetoelectricity

Although the crystal symmetry *R*-3 of CoTe$_6$O$_{13}$ is centrosymmetric, the antiferromagnetic magnetic structure breaks spatial inversion in the magnetic lattice and thus can lead to coupling between the magnetization and induced electric polarization. Temperature- and field-dependent dielectric permittivity measurements were therefore performed in *E*//*H* and *E*⊥*H* geometries on a polycrystalline pellet of CoTe$_6$O$_{13}$. As shown in Fig. 6a, b, no dielectric anomaly is observed in zero field, but clear peaks at $T_N$ appear after applying fields either parallel or perpendicular to the electric field. This behavior has been widely observed in linear magnetoelectric materials. Consistently, the pyroelectric current displayed in Fig. 6c, d shows a clear peak at $T_N$ in fields either parallel or perpendicular to the current, indicating the development of electric polarization induced by the magnetism. The value of electric polarization is obtained by integrating the current density with time, which is around 140 μC m$^{-2}$ in 9 T field parallel to the current and around 280 μC m$^{-2}$ in 9 T field perpendicular to the current. Moreover, the off-diagonal pyroelectric current in Fig. 6d shows a clear shoulder peak feature below $T_N$. This may be related with a local magnetic structure, which is also observed in the toroidal magnetoelectric material LiNiPO$_4$[23]. As shown in Fig. 7a, b, diagonal and off-diagonal magnetoelectricity is demonstrated by continuously collecting the current signal during sweeping the magnetic field from 9 T to −9 T to 9 T for three cycles at 1.8 K, parallel to the current and perpendicular to the current, respectively. Both diagonal and off-diagonal magnetoelectric current signals are observed. Note that the magnetoelectric current has an almost constant value over the full −9 T to +9 T range, but the current direction depends on the sweeping direction of the magnetic field. This is a signature of linear magnetoelectricity. Fig. 7c and d display the applied magnetic field and obtained diagonal and off-diagonal electric polarization as a function of time, which clearly demonstrates the presence of a linear magnetoelectricity in the −9 T to 9 T range. The measured diagonal and off-diagonal magnetoelectric coefficients ($dP/dH$) are around 27.9 ps/m and 41.2 ps/m, respectively. As shown in Fig. S6, a single crystal plate was prepared to collect the polarization signal perpendicular to the *c* axis, and magnetic field was applied perpendicular to the *c* axis and the crystal plate's normal direction. This geometry is to measure only the $\sigma_{12}$ element in the magnetoelectric tensor. The measurement results give $\sigma_{12}$ = 49.5 ps/m, similar to the value obtained on the polycrystalline sample, which confirms that the off-diagonal magnetoelectric response in CoTe$_6$O$_{13}$ mostly comes from the $\sigma_{12}$ element. As displayed in Fig. S7, the magnetoelectric current loops collected at 2 K, 5 K, and 10 K are almost identical and show linear magnetoelectricity within the −9 T to 9 T range, while anomalies in the magnetoelectric response appear at around ±7 T in the 15 K data, and the polarization in the high-field state is reduced. Those anomalies coincide with the metamagnetic transition shown in Fig. 2c, and this dataset implies that the metamagnetic transition suppresses the out-of-plane toroidal moment.

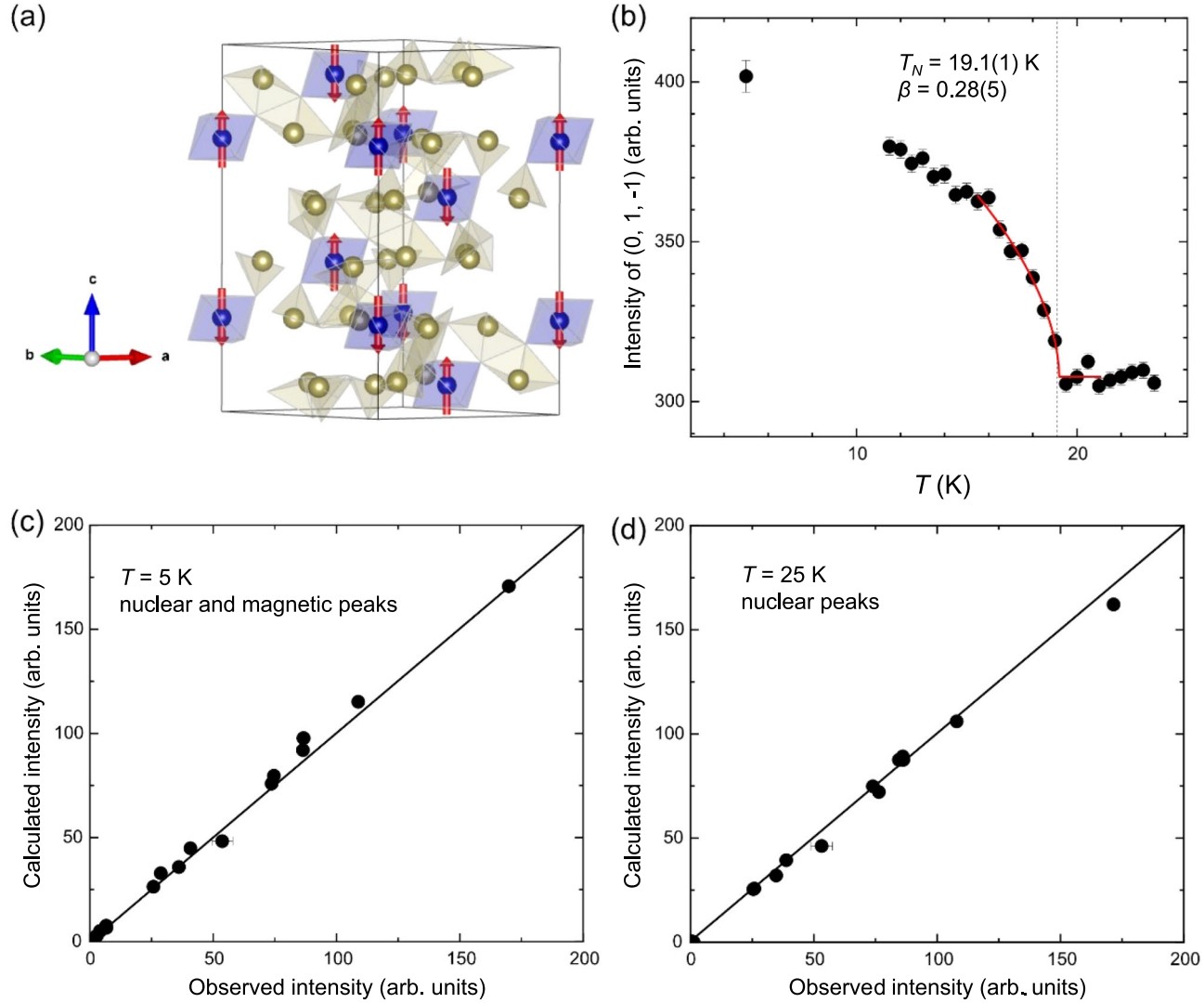

**Fig. 5 | Single-crystal neutron diffraction. a** The magnetic structure below $T_N$ determined by single-crystal neutron diffraction. **b** Temperature evolution of (0, 1, -1) magnetic peak intensity (black dots). the red line is a power law fitting with the formula $I-I_0 = A(1 - T/T_N)^{2\beta}$. The point at 5 K is normalized from fitting the rocking curve scans of (0, 1, -1) peak at 5 K, 15 K and 25 K. **c, d** Calculated intensity vs. observed intensity at 5 K (nuclear and magnetic peaks) and 25 K (nuclear peaks only), respectively. The error bars in this figure display the standard deviations of the intensity.

## Discussion

Off-diagonal linear magnetoelectricity stems from magnetic toroidicity, which requires spatial inversion and time reversal symmetries both to be broken. The magnetic point group -3′ of CoTe$_6$O$_{13}$ satisfies the symmetry requirements for off-diagonal magnetoelectricity. Researchers have proposed that collinear antiferromagnetic spins can create toroidicity via a dimer distortion, as illustrated in Fig. 8a[40,41]. However, this scenario is not enough to explain the observed off-diagonal magnetoelectricity in CoTe$_6$O$_{13}$, since the -3′ point group is

supposed to have the magnetoelectric tensor $\sigma = \begin{bmatrix} \sigma_{11} & \sigma_{12} & 0 \\ -\sigma_{12} & \sigma_{11} & 0 \\ 0 & 0 & \sigma_{33} \end{bmatrix}$

based on Neumann's principle. The non-zero $\sigma_{12}$ element requires a toroidal moment along the $c$ axis, but the dimerization of spins along the $c$ axis in CoTe$_6$O$_{13}$ can only possibly create a toroidal moment perpendicular to the $c$ axis, not along the $c$ axis.

As shown in the left panel of Fig. 8b, if we introduce mirror symmetries (mirror planes parallel to $c$ axis) into the Co$^{2+}$ triangular lattice, i.e., removing the octahedron distortions, and keeping the Co$^{2+}$ sublattice and spins unchanged, the resultant magnetic point group

becomes -3′m′ (the same as Cr$_2$O$_3$), which has magnetoelectric tensor

$\sigma = \begin{bmatrix} \sigma_{11} & 0 & 0 \\ 0 & \sigma_{11} & 0 \\ 0 & 0 & \sigma_{33} \end{bmatrix}$. This fact implies that the structural ferro-rotation

distortion that breaks mirror symmetries is the essence of having off-diagonal magnetoelectricity in CoTe$_6$O$_{13}$.

In the -3′m′ case (Fig. 8b), though mirror symmetries are introduced into the lattice, the whole spin texture of the ground state is still mirror-broken (left panel of Fig. 8b), since spins along the $c$ axis break mirror planes parallel to the $c$ axis anyway. On the contrary, in the toroidal configuration (right panel of Fig. 8b), the in-plane spins perpendicular to the mirror planes do not break the mirror symmetries, and the whole spin texture regains those mirror symmetries with a magnetic point group -3′m, which is different to the ground state. Note that the spins in -3′m′ are allowed to have local canting moments parallel to the broken mirror planes (middle panel of Fig. 8b), but this configuration has zero toroidicity. In conclusion, -3′m′ has symmetries that are not compatible with magnetic toroidicity, and thus will not exhibit an off-diagonal magnetoelectric response.

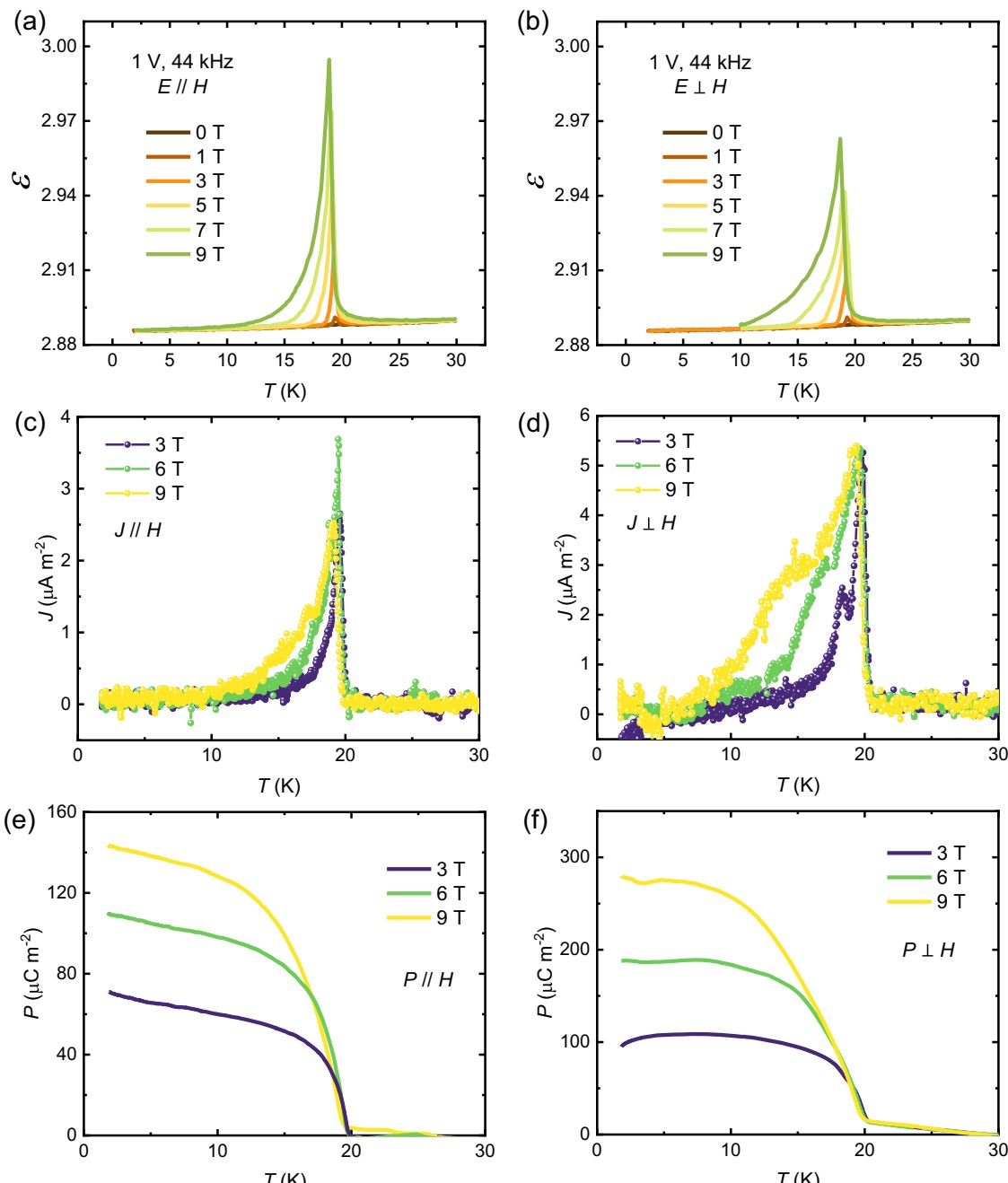

**Fig. 6 | Dielectric and pyroelectric anomaly at $T_N$.** **a**, **b** The dielectric permittivity ($\varepsilon$) measured in magnetic fields parallel and perpendicular to the electric fields, respectively. The magnitudes of fields are displayed by different curve colors shown in legend. **c**, **d** The pyroelectric current density ($J$) measured in magnetic fields parallel and perpendicular to the current, respectively. Blue, green and yellow spheres and curves denote the data measured at 3 T, 6 T, and 9 T fields, respectively. **e**, **f** The electric polarization ($P$) vs. temperature obtained by integrating the current density with time in magnetic fields parallel and perpendicular to the current, respectively. Blue, green and yellow curves denote the data measured at 3 T, 6 T, and 9 T fields, respectively.

The situation in -3' is different. As shown in the right panel of Fig. 8c, despite the fact that the in-plane spins do not break mirror planes parallel to the $c$ axis, the mirror symmetries of the whole texture are always broken due to the ferro-rotation structural distortion. Both the collinear spin texture (Fig. 8c left panel) and the toroidal spin texture (Fig. 8c right panel) only have three-fold rotational symmetry along $c$ and a combination of spatial inversion and time reversal symmetry, without any additional symmetry elements such as mirror planes. Therefore, those two textures are equivalent in terms of symmetry operations, which suggests that the canting for an out-of-plane toroidal moment is allowed in -3'. Note that we did not observe any superlattice peaks allowing ordered canted spins in the neutron

diffraction below $T_N$, which suggests that the spin canting model in $CoTe_6O_{13}$ is an allowed local magnetic mode and does not form a long-range ordering without applied stimulation. Then, the observed broad shoulder peak features in $d\chi/dT$ (Fig. 2b) and pyroelectric current (Fig. 6d) below $T_N$ might be related to the hidden local magnetic symmetry breaking. Although the canting moments do not have long-range ordering, a macroscopic observable magnetic toroidicity can still be stabilized by applying poling $E$ and $H$ during cooling (see Methods section). Similarly, magnetic toroidicity and off-diagonal magnetoelectric effect have been realized in a spin glass through a proper $E$, $H$ poling[42]. We notice that the reported magnetic structure of a $Yb_3Pt_4$ compound (same space group $R$-3, same magnetic point

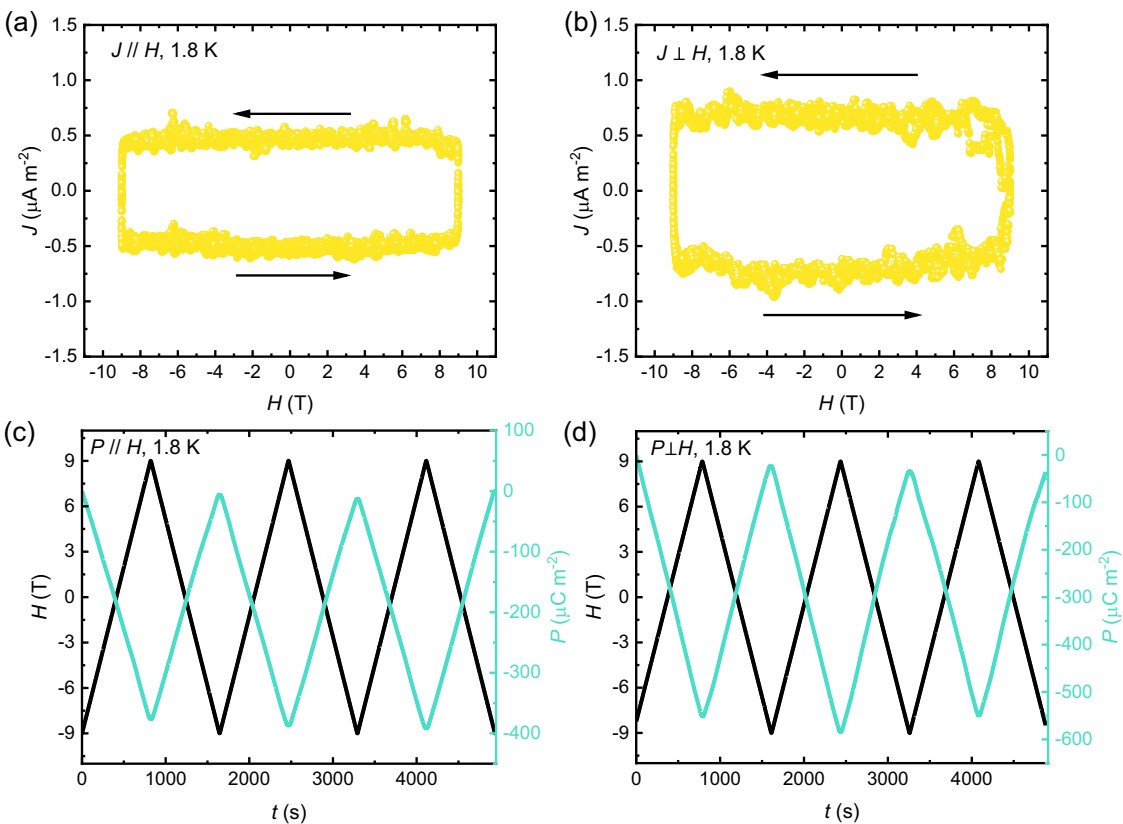

**Fig. 7 | Magnetoelectric response. a**, **b** The magnetoelectric current density (*J*) at 1.8 K measured with magnetic fields parallel and perpendicular to the current, respectively. The field sweeping includes three −9 T to 9 T to −9 T cycles. **c**, **d** The sweeping magnetic fields (*H*) and electric polarizations (*P*) obtained from (**a**) and (**b**) as a function of time (*t*).

group -3′, and same **k** = (0,0,0)) consists of an out-of-plane A-type antiferromagnetism plus an in-plane 120° vortex from canting[43], which confirms that -3′ symmetry allows the local canting and out-of-plane toroidal moments. The only difference is that the in-plane moments form a 120° long-range order in Yb$_3$Pt$_4$ due to Yb site splitting (not at a high-symmetry site as Co in CoTe$_6$O$_{13}$). To probe the possible local symmetry breaking below $T_N$, NPD patterns were collected at intervals of 1 K during a ramping from 8 K to 40 K, and the obtained temperature contour plots of (1 1 3), (0 2 1), and (1 0 1) peaks are displayed in Fig. S4c to S4e. For the nuclear plus magnetic peak (0 2 1), the temperature-dependent full width at half maximum (FWHM) obtained from the Gaussian fitting of the peak profiles unveils a slight broadening tendency below $T_N$ (Fig. S4g), while another nuclear plus magnetic peak (1 1 3) does not show systematic change of FWHM from 8 K to 40 K (Fig. S4f). So far, the data in the current work cannot be taken as conclusive evidence of local canting. In future studies, local-sensitive neutron technique, e.g., magnetic pair distribution function (*mPDF*) analysis could describe the local magnetic symmetry of CoTe$_6$O$_{13}$ in a more decisive manner[44]. Nevertheless, the phenomenological picture of the emergence of a toroidal moment in CoTe$_6$O$_{13}$ can be depicted as the diagonal magnetoelectric magnetic sublattice mapping the electric dipole vortices (ferro-rotation) into spin vortices.

Table 2 summarizes the linear magnetoelectric coefficients and off-diagonal response of some reported magnetoelectric materials. In fact, most of them show a collinear spin structure except BaCoSiO$_4$. The off-diagonal linear magnetoelectric coefficient that we observe, 41.2 ps/m, for the CoTe$_6$O$_{13}$ polycrystalline sample is only smaller than that of TbPO$_4$[18]. But the Neel temperature of TbPO$_4$ is only 2.38 K, which is much lower than the Neel temperature of CoTe$_6$O$_{13}$ (19.5 K). The polar ferri-magnet Fe$_2$Mo$_3$O$_8$ is reported to exhibit a huge diagonal magnetoelectric coefficient ~ 5700 ps/m, but this only happens over an extremely narrow field regime corresponding to the meta-magnetic transition[45]. In contrast, CoTe$_6$O$_{13}$ has linear magnetoelectric responses over the full -9 T to 9 T range with a fairly constant magnetoelectric coefficient. Overall, CoTe$_6$O$_{13}$ exhibits the largest off-diagonal linear magnetoelectric coefficient among all reported transition metal magnets to the best of our knowledge. Since the existence of magnetic toroidicity in CoTe$_6$O$_{13}$ is enabled by the structural ferro-rotation distortion (Fig. 8), we speculate that the large off-diagonal magnetoelectricity in CoTe$_6$O$_{13}$ is related to its large ferro-rotation distortion magnitude (Fig. S1). Moreover, unlike some previous systems having a net ferromagnetic/ferrimagnetic moment coupled to the toroidal moment, such as BaCoSiO$_4$ and LiCoPO$_4$[20,21], no hint of net ferromagnetic/ferrimagnetic moment is detected in CoTe$_6$O$_{13}$, and a butterfly-shaped hysteresis does not exist in the magnetoelectric response in a consistent manner, which indicates that the toroidal moment in CoTe$_6$O$_{13}$ stays mono-domain over the −9 T to +9 T range at 1.8 K, and that the direction of toroidal moment is determined by the applied *E*, *H* poling during cooling. Therefore, CoTe$_6$O$_{13}$ is a unique venue for studying the physics of magnetic toroidal ordering without interlocked ferromagnetic net moments.

In conclusion, though a straightforward picture of magnetic toroidicity is a spin vortex, a more general type of magnetic toroidicity in real materials is one of the collinear spins plus broken crystallographic symmetries. Moreover, collinear spins may generate a toroidal moment perpendicular to them through dimerization or a toroidal moment parallel to them through canting, depending on the crystallographic symmetries. Previously established pictures mostly focus on the spin dimerization situation, but our experimental findings in CoTe$_6$O$_{13}$ strongly suggest that the canting of collinear spins allowed by broken mirror symmetries is also a feasible path toward magnetic toroidicity and off-diagonal magnetoelectricity. In addition, these two

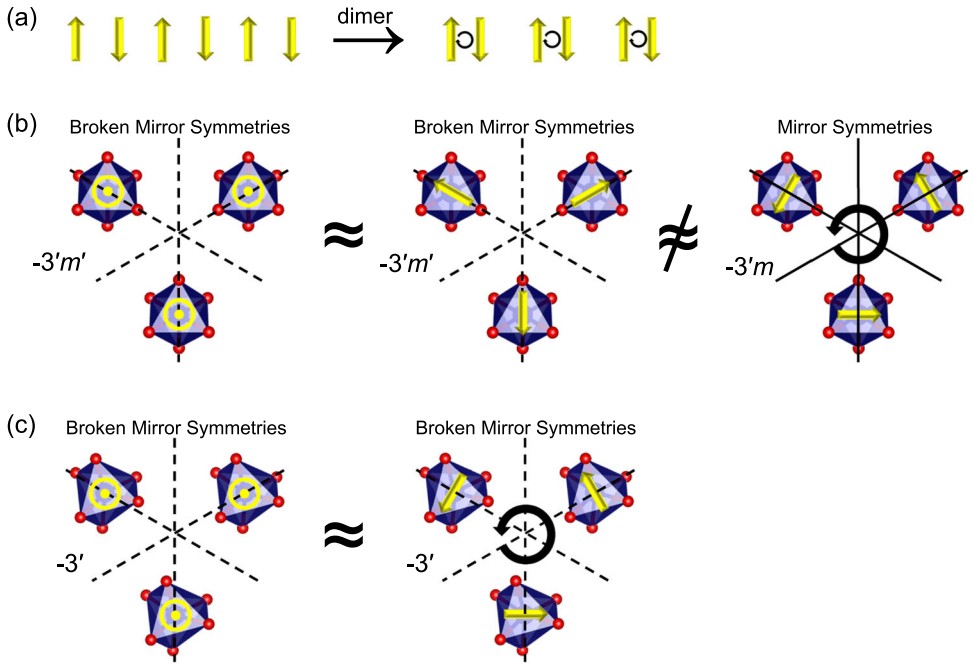

**Fig. 8 | Symmetry analysis.** Schematic diagram of symmetry-operational similarities. The dashed lines denote broken mirror planes, and the solid lines represent unbroken mirror planes. **a** Collinear spins create toroidal moments perpendicular to them through dimerization. **b** Out-of-plane spins in a triangular lattice without ferro-rotation have symmetry similarity to a monopole (middle), but distinct symmetry to a toroidal moment (right). **c** Out-of-plane spins in a triangular lattice with ferro-rotation have symmetry similarity with an out-of-plane toroidal moment that consists of in-plane spin components.

## Table 2 | Summary of examplary linear magnetoelectric materials

| Compound | Diagonal linear ME coefficient (ps/m) | Off-diagonal linear ME coefficient (ps/m) | Magnetic structure | Expt. temperature | References |
|---|---|---|---|---|---|
| $CoTe_6O_{13}$ | 27.9 | 41.2 | collinear | 1.8 K | this work |
| $BaCoSiO_4$ | 5.42 | 1.03 | 120° vortex | 1.8 K | 20 |
| $Cr_2O_3$ | 0.23 | 0.73[a] | collinear | 4.2 K | 64 |
| $TbPO_4$ | 0 | 730 | collinear | 1.5 K | 18 |
| $LiCoPO_4$ | 0 | 30.6 | collinear | 4.2 K | 21 |
| $LiNiPO_4$ | 0 | 1.5 | collinear | 4 K | 23 |
| $BaNi_2(PO_4)_2$ | 1.67 | 1.67 | collinear | 5 K | 46 |
| $CaMnGe_2O_6$ | unknown | 0.06 | collinear | 2 K | 24 |
| $CoSe_2O_5$ | 0 | 21.3 | collinear | 4 K | 65 |

[a]The off-diagonal ME coefficient of $Cr_2O_3$ was reported in the high-field spin-flopped phase. The ground state having magnetic point group -3′m′ does not allow off-diagonal ME effect.

mechanisms should co-exist in collinear magnets with magnetic point group -**1**′ such as $BaNi_2(PO_4)_2$[46], $Fe_4Nb_2O_9$[47,48], and $MnPSe_3$, since they are supposed to have all nine non-zero elements in their linear magnetoelectric tensor. Based on the reported magnetic symmetries, the similar physics picture of toroidal moment parallel to collinear spins is also expected to exist in compounds such as $MnTiO_3$ (-3′)[49], $GaFeO_3$ (*m′m2′*)[50], $TlFe_{1.6}Se_2$ (4/*m′*)[51], $K_{0.8}Fe_{1.8}Se_2$ (4/*m′*)[52], $CsCoF_4$ (-4′)[53], $KOsO_4$ (4′/*m′*)[54], and $KRuO_4$ (4′/*m′*)[55], and the local magnetic symmetry breaking of these materials is worth testing in forthcoming experiments. Note that, though the symmetry analysis in the present work suggests the existence of toroidal moments by locally canted collinear spins, it cannot give quantitative information on the magnitude of the induced toroidal moment. Therefore, a rigorous theoretical calculation would be helpful for a deeper understanding of the observed large off-diagonal magnetoelectricity. Recently, researchers have proposed the concepts of altermagnetism[56,57] or Trompe L'oeil ferromagnetism[58] to describe a type of antiferromagnetism that behaves like ferromagnetism under certain external perturbations. In a similar manner, the present work describes a type of collinear antiferromagnetism that

behaves like a ferro-toroidicity, and it may initiate the further theoretical and experimental study of emergent physical phenomena predicted by symmetry analysis. In addition, the observed large effective moment in the paramagnetic regime of $CoTe_6O_{13}$ indicates potential applications in such areas as rare-earth-free super-paramagnets. Also, experimental demonstrations of more functional behaviors arising from the magnetic toroidicity in $CoTe_6O_{13}$, such as a non-reciprocal directional dichroism effect, are of great interest.

## Methods

Polycrystalline $CoTe_6O_{13}$ was synthesized via a conventional solid-state reaction method. $CoCO_3$ (Alfa Aesar, 99%) and $TeO_2$ (Alfa Aesar, 99.99%) powders in a molar ratio 1:6 were mixed, pelletized, and sealed in a thick-walled evacuated quartz tube. The pellet characterized was sintered at 600°C for 48 hours with one intermediate grinding. The isostructural non-magnetic compound $MgTe_6O_{13}$ was synthesized by a similar method but using a higher sintering temperature 630°C, with MgO (Alfa Aesar, 99.95%, 900°C overnight baked) and $TeO_2$ powders in a molar ratio 1:6 as the

starting materials. The XRD pattern and Rietveld refinement result of $MgTe_6O_{13}$ are shown in Fig. S2c.

$CoTe_6O_{13}$ single crystals were grown via a chemical vapor transport method. 0.5 gram of polycrystalline $CoTe_6O_{13}$ powder was sealed in a 10-cm quartz tube. 0.08 g $TeCl_4$ was added as the transport agent. The tube was placed horizontally in a multi-zone tube furnace. The hot zone with starting materials was kept at 620°C, and the cold zone was kept at 560°C. After two weeks, crystals with a typical mass of several milligrams were collected at the cold zone.

The crystal structure was determined using a Bruker single crystal x-ray diffractometer. The structure was refined using the SHELXTL Software Package[59,60], in the centrosymmetric space group $R$-3. The crystallographic data are listed in the Supplementary Information.

The magnetism and heat capacity measurements were carried out in a Quantum Design Dynacool PPMS-9. $\chi(T)$ was measured with 1000 Oe applied magnetic field. The dielectric permittivity was measured using a QuadTech 7600 LCR meter externally connected to the PPMS sample probe. The electrodes were made of silver epoxy. The pyroelectric and magnetoelectric currents were collected using a Keithley 617 source meter externally connected to the PPMS sample probe. The samples were cooled down to 1.8 K with a 100 kV/cm DC electric field (applied by a Keithley 2400 source meter) and a 3 T magnetic field applied to induce a mono magnetic domain state. All magnetoelectric current measurements were performed via 200 Oe/s magnetic field sweeping at fixed temperatures. The heating rate for all pyroelectric measurements was 5 K/min. The symmetry analysis of magnetoelectric tensors used the *Bilbao Crystallography Server*[61].

The single crystal neutron diffraction was performed at HB-3A DEMAND[62] instrument at High Flux Isotopy Reactor at Oak Ridge National Laboratory. A piece of single crystal sample was cooled down to 5 K in a closed cycle refrigerator (CCR). The sample was measured in the four-circle mode. The neutron wavelength used was 1.542 Å from the bent Si-220 monochromator.

The neutron powder diffraction on the powder sample was conducted at time-of-flight powder diffractometer POWGEN. A Powgen automatic changer (PAC) was adopted to cover the temperature region 8 K-300 K. The neutron frame with center wavelength of 1.5 Å was used for the long data collection at 8 K, 14 and 40 K. To track the temperature dependence of the low-Q nuclear and/or magnetic peaks, another neutron frame with center wavelength of 2.665 Å and higher resolution was used for the data collection while warming the sample from 8 K to 40 K with a slow ramping rate of 0.4 K/m. The symmetry analysis used *Bilbao Crystallographic Server*[61]. The magnetic structure refinement used *FULLPROF* software suite[63].

Scanning electron microscopy (SEM) images and Energy-dispersive x-ray spectroscopy (EDS) were collected using a Quanta 200 FEG Environmental-SEM on the as-grown surface of a $CoTe_6O_{13}$ crystal.

The X-ray photoelectron spectroscopy (XPS) data were collected using the Thermo Scientific K-Alpha XPS UPS system on the as-grown surface of a $CoTe_6O_{13}$ crystal. Each spectrum was averaged with 30 frames of 30 seconds dwelling time each, under $10^{-8}$ torr vacuum environment.

## Data availability

The x-ray diffraction, magnetization, heat capacity, dielectric, pyroelectric, and magnetoelectric data generated in this study are provided in the Source Data file. Source data are provided with this paper.

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

## Acknowledgements

X.X., D.N., C.Y., and R.J.C. acknowledge the NSF-MRSEC Grant No. DMR-2011750 for supporting the work at Princeton University. Y.H. and H.C. acknowledge the support from the US Department of Energy (DOE), Office of Science, Office of Basic Energy Sciences, Early Career Research Program Award KC0402020, under Contract No. DE-AC05-00OR22725. This research used resources at the High Flux Isotope Reactor and the Spallation Neutron Source, the DOE Office of Science User Facility operated by ORNL.

## Author contributions

X.X. conceived the CoTe$_6$O$_{13}$ project. R.J.C. supervised this work. X.X. grew the samples, measured the x-ray diffraction, magnetization, heat capacity, dielectric and magnetoelectric data and performed analysis. D.N. and C.Y. conducted chemical characterization experiments and analysis on the studied samples. Y.H. and H.C. performed neutron single-crystal diffraction experiments and data analysis. Q.Z. performed neutron powder diffraction experiments and data analysis. S.P. and X.D. performed theoretical modeling and calculations of the Co$^{2+}$ ion magnetism. X.X., Y.H., S.P., Q.Z., H.C., and R.J.C. wrote the paper with comments from all the authors.

## Competing interests

The authors declare no competing interests.
