## [Peer Review File · Nature Communications]

Reviewers' Comments:

Reviewer #1:

Remarks to the Author:

The authors reported the magnetic toroidicity in an antiferromagnet CoTe₆O₁₃ ordered ~ 20 K, exhibiting a linear magnetoelectric effect below the ordering temperature. Symmetry analyses suggested that a deduced ferro-rotation-type structural order enables the off-diagonal magnetoelectricity. In this case, the CoTe₆O₁₃ could be a new multiferroic material for further study. The authors presented an interesting work that can be potentially published on Nature Communications. However, before it can be accepted, further discussion is required to clarify some unclear points.

1, In the magnetism section, the author mentioned orbital contribution to the large effective magnetic moment, which is quite important in this research. To further support this point, theoretical simulations are recommended.

2, In the neutron scattering session, I am curious about the experiment set-up, the "four-circle mode", therefore, I believe more details are required e.g., the sample alignment details, the obtained diffraction patterns and refinement details. The temperature dependent intensity plot shown in Fig. 5b only demonstrates the intensity of one Bragg/magnetic peak (01-1). There're other allowed magnetic peaks may contain more information, e.g., the broaden peak feature in Fig. 2b. Moreover, the error bar of 5-K point in Fig. 5b is wired, which is significantly larger than other data points. What is the intensity here, monitor counts; and why the error bar is so different, considering the intensity difference is not great? The temperature dependent powder diffraction patterns tell more information than single crystal diffraction in solving magnetic structure.

3, In the magnetoelectric session, a broaden feature is shown in Fig. 6d, which seems like the feature in Fig. 2b. However, I can't see any clear "hint" in Fig. 5b. This is very important, since the authors also believe the "possible short-range magnetic interaction" here is coupled with the magnetoelectric effect. A high-resolution NPD experiment at low temperatures will solve all the magnetic structure related problems, and present a clear picture in this compound.

Some minor points:

- Define "ferro-rotation".
- Line 157, missing "+" in Co²⁺.
- Please use italics for the symbols of physical quantities.

The overall science in this manuscript is fantastic. However, I believe the magnetic structure and low-temperature crystal structure are very important to the claims in this work. If the authors present more convinced experiment results in a revised version, I am happy to suggest its publication in Nature Communications.

Reviewer #2:

Remarks to the Author:

Referee report on 'Large Off-diagonal Magnetoelectricity in a Triangular Co²⁺-based Collinear Antiferromagnet'

I have reviewed this manuscript.

Xu et al. used a comprehensive set of experimental techniques to characterize poly- and single-crystalline samples of CoTe₆O₁₃. They described their observations in detail, complemented by some symmetry analysis.

The highlight is the observation of a large off-diagonal magnetoelectric effect in a long-ranged collinear spin structure, which is argued to be related to magnetic toroidicity.

Studies of this type are highly appreciated, as they assess an exotic phenomenon in physics – off-diagonal magnetoelectricity – via different angles, let alone the great potential held by multiferroics in device applications. The manuscript overall is well written and structured.

However, I have also identified items that need to be taken care of before I can recommend for publication. My overall judgements on these items are:

- They need to deliver more clear information.
- They need to be discussed in a more rigorous manner.

As a result, my recommendation is that the authors need to carry out a major revision to the current manuscript to address my concerns, and to meet the publication criteria of Nature Communications.

I append my specific comments below, hoping that the authors can consider all of them when improving their manuscript.

1. In the abstract, the authors claim that 'Our results establish a symmetry criterion for a new mechanism of magnetic toroidicity, induced by collinear spins parallel to the toroidal moment'. In the second paragraph of the main manuscript, they also state that the origin of magnetic toroidicity in collinear spin systems is not straightforward. Based on these, it is natural for me to expect that they can shed new light on this puzzle in the current work. Unfortunately, the relevant discussions on symmetry are not very clear. For example, is canting, which is not necessarily related to the collinear structure, critical for generating toroidicity? I will comment on canting in more detail below.

2. Fig. 1a displays a Le Bail fitting on the powder XRD data, i.e. profile matching. This is not a standard approach when fully solving a crystallographic structure. Can they present the Rietveld fitting instead? Or, if they choose to rely on the single crystal XRD refinement, they should plot those data accordingly, e.g. observed vs calculated intensities.

3. Fig. 2b, I am not convinced that one can directly/strongly argue short-range ordering based on the derivative of magnetic susceptibility. Is there any supporting literature, which used a similar approach to deliver this message, where the short-range ordering is also directly observed by, for example, neutron scattering?

4. The authors also compared their magnetization data with those published. This was summarized in Fig. S4. In the caption, they state that they 'guessed' the magnetic field used in the previous published work. This should not be encouraged, as in peer reviewed publications, it is not appropriate to cite guessed measurement conditions. I suggest removing this comparison.

5. Heat capacity measurements are used to support the effective spin at low temperatures and broad feature that might point to magnetic short-range ordering. This goes beyond simply determining the transition temperature. In practice, analysis of this kind requires a careful calibration. Subtraction of nonmagnetic contributions in the heat capacity data is critical for correctly reproducing the magnetic contributions in absolute unit, as well as the shape of its temperature dependence. In my opinion, these data provide the only hint in their work supporting that there is some short-range ordering below ~ 20 K at low field, which is buried in the noise of their neutron measurements. I would suggest the authors to carry out a separate nonmagnetic background measurement, e.g. on NMTe6O13, where NM is a nonmagnetic element. This is important because their discussion later is largely relying on the assumption that there is a local canting. Moreover, this also can further validate if the low temperature effective spin is indeed $3/2$. As the authors stated, in many other cases, Co ions in similar crystal field environment has an effective spin of $1/2$ at low temperatures.

6. Significant amount of information is missing in the presentation of their neutron diffraction data analysis.

a. For the audience, it is not straightforward to get the symmetry elements out by simply quoting the BNS symbols. Since the symmetry is important for understanding the magnetoelectric effect, please present a detailed symmetry analysis, e.g. irreducible representations and basis vectors.

b. How many fitting parameters are there in the magnetic structure refinement? What are they? Is multi-domain configuration or spin canting already allowed by symmetry?

c. In Fig. 5a, how many points have non-negligible magnetic contributions? Perhaps, it is better to make a table, separately listing the nuclear and magnetic contributions at these locations, since $k = (0, 0, 0)$.

7. The critical scaling analysis in Fig. 5b is incorrect, as this should only be applied to a narrow region near T_N , given that the data density there is sufficient.

8. In Magnetoelectricity, the authors state 'Moreover, the off-diagonal pyroelectric current in Fig. 6d shows a clear shoulder peak feature below T_N . This may be related with a short-range magnetic ordering, which is also observed in the toroidal magnetoelectric material LiNiPO_4 '. Since

these measurements were performed in a sizable magnetic field, does the incomplete metamagnetic transition shown in Fig. 2c play a role in generating the short-range order?

9. In Magnetoelectricity, the authors state 'Neutron diffraction (Fig. 5b) and magnetic susceptibility (Fig. 2b) data also show hints of that transition'. I cannot see such hints from the neutron diffraction data presented. They will need to explain this statement. My comments on the susceptibility data can be found in Point 3 above.

10. The magnetoelectric current loop measurements were done at 1.8 K (Fig. 7). If this effect is coupled to the magnetic structure, why aren't these measurements sensitive to the metamagnetic transition in Fig. 2c?

11. In the Discussion, the authors say 'As shown in the left panel of Fig. 8, if we introduce mirror symmetries (mirror planes parallel to c axis) into the Co²⁺ triangular lattice, i.e., removing the octahedron distortions, and keeping the magnetic structure unchanged...'. This assumption does not sound solid to me. From symmetry point of view, when you change the nuclear symmetry, the magnetic symmetry may also change. What they described following this assumption certainly is one plausible scenario, but the limitations of this assumption need to be addressed.

12. 'Therefore, collinear spins along c in the -3' point group show symmetry operation similarity to a toroidal moment along c formed by in-plane head-to-tail spin components.' I cannot fully catch the point of this statement. If the authors believe that there is a connection between the two cases. A rigorous discussion is required.

13. In the last part, the authors discussed short-ranged canting, and linked the system to other compounds reported in the literature. However, in my opinion, no clear evidence supporting canting is seen from their data. The first question one could ask is - is canting allowed by $k = (0, 0, 0)$? If not, a second propagation vector is required. Then the magnetic moment refinement presented above is not as accurate. Moreover, can short-ranged canting lead to such large off-diagonal magnetoelectricity? The effect reported in Ref. 42 is smaller.

14. Side comment: ac susceptibility measurements could be helpful to support their claim of short-range magnetic ordering.

End of report.

All changes in the manuscript are highlighted in yellow.

Reviewer #1

The authors reported the magnetic toroidicity in an antiferromagnet CoTe₆O₁₃ ordered ~ 20 K, exhibiting a linear magnetoelectric effect below the ordering temperature. Symmetry analyses suggested that a deduced ferro-rotation-type structural order enables the off-diagonal magnetoelectricity. In this case, the CoTe₆O₁₃ could be a new multiferroic material for further study. The authors presented an interesting work that can be potentially published on Nature Communications. However, before it can be accepted, further discussion is required to clarify some unclear points.

Reply:

We appreciate the reviewer's nice summary and positive comments on our manuscript. Thank you! We have made changes to our manuscript to address your concerns and provided more evidence to strengthen our conclusions. Kindly refer to the details provided below. Thank you.

1, In the magnetism section, the author mentioned orbital contribution to the large effective magnetic moment, which is quite important in this research. To further support this point, theoretical simulations are recommended.

Reply:

Thanks for the good suggestion. We have performed theoretical calculations on the effective moment of Co²⁺ ions in CoTe₆O₁₃. The related discussion is shown in section iii and SI section i. in the revised manuscript. The calculated *c* axis moment is 6.98 μ_B with 28% orbital contribution, which is close to the experimental result. The calculated *ab* plane moment is much smaller than the experimental result, and we believe the reason is that the CW law does not apply in the 200 K to 300 K range along *ab*, because the *ab* plane magnetic interactions are strong and not negligible in that temperature range, evidenced by the large |T_{CW}| along *ab*.

2, In the neutron scattering session, I am curious about the experiment set-up, the “four-circle mode”, therefore, I believe more details are required e.g., the sample alignment details, the obtained diffraction patterns and refinement details. The temperature dependent intensity plot shown in Fig. 5b only demonstrates the intensity of one Bragg/magnetic peak (01-1). There're other allowed magnetic peaks may contain more information, e.g., the broaden peak feature in Fig, 2b. Moreover, the error bar of 5-K point in Fig. 5b is wired, which is significantly larger than other data points. What is the intensity here, monitor counts; and why the error bar is so different, considering the intensity difference is not great? The temperature dependent powder diffraction patterns tell more information than single crystal diffraction in solving magnetic structure.

Reply:

Thanks for the comment. Here we provide more details in the four-circle experiment setup: The single-crystal sample is aligned proximate to (HHL) scattering plane. In four-circle mode, the sample is mounted on the four-circle goniometers, which allows -180~180-degree sample-rotation, -40~50-degree vertical rotation, and 0~45-degree horizontal rotation. This setup provides access to the whole reciprocal space. More details for the four-circle measurement and the instrumentation specification can be found in ref. 29. In the peak measurements, the peak is put to the detector center,

then a short scan (rocking curve) was measured by ± 1 -degree horizontal rotation using 0.1-degree step. 23 peaks were measured at both 5 K and 25 K, of which 9 peaks have non-negligible magnetic intensity.

The different error bar in the temperature dependence is because the point at 5 K was extracted using fittings of the rocking curve scans. The rocking curves measured at 5 K and 15 K were fitted using a Gaussian function. The fitted peak intensities were normalized to the temperature dependence scan to give the point at 5 K. The fitted peak heights are 129.6 ± 6.0 at 5 K and 102.1 ± 4.5 at 15 K. An explanation has been added to the figure 5 caption: “The point at 5 K is normalized from fitting the rocking curve scans of (0, 1, -1) peak at 5 K, 15 K and 25 K.” (line 466 - 467)

Figure R1 | Rocking curves of (0, 1, -1) peak measured at 5 K, 15 K and 25 K.

We have carried out a neutron powder diffraction (NPD) on 5 grams of $\text{CoTe}_6\text{O}_{13}$ powder sample. Three long scans were collected at 8 K, 14 K, and 40 K, and quick scans with 1 K interval are collected within a ramping from 8 K to 40 K.

The patterns can be well refined using the $R\text{-}3'$ model (new Fig. S5). The peak indices with non-negligible magnetic contributions are summarized in Table SVI. All indices are consistent with the reflection condition for $R\text{-}3'$, which is $2h + k + l = 3n$, and no violated diffractions are observed. Therefore, the NPD results further support the magnetic structure obtained from single-crystal neutron diffraction.

Figure S5. Refinement results of neutron powder diffraction patterns at (a) 40 K, (b) 14 K, and (c) 8 K using the $R-3'$ model. The obtained magnetic moment (m_z) of Co^{2+} is $3.81 \mu_B$ at 14 K and $4.43 \mu_B$ at 8 K.

Relevant discussion is added into the revised manuscript:

“To ascertain the adopted magnetic model, neutron powder diffraction (NPD) experiments are performed on an around 5-gram polycrystalline sample of $\text{CoTe}_6\text{O}_{13}$ at three temperatures, i.e., 40 K, 14 K and 8 K. As shown in Fig. S4, additional diffraction intensities below T_N demonstrate magnetic diffractions from the ordered magnetic state. Table SVI summarizes the Bragg indices of observed magnetic peaks, and all reflections can be indexed with the crystallographic unit cell, which is consistent with a wave vector $\mathbf{k} = (0, 0, 0)$. Using the $R-3'$ (BNS #148.17) model, the NPD patterns can be well fitted, and the refinement results are displayed in Fig. S5. The obtained magnetic moment (m_z) of Co^{2+} is $3.81(4) \mu_B$ at 14 K and $4.43(4) \mu_B$ at 8 K, which is quite close to the value obtained from the single-crystal neutron diffraction. Overall, the single-crystal and powder neutron diffraction results consistently manifest the $\mathbf{k} = (0, 0, 0)$ and $-3'$ magnetic ordering.” (line 271 - 280)

Temperature-dependent contour of NPD results are displayed in new Fig. S4. Please refer to the response in the subsequent question for a detailed discussion of these data.

3, In the magnetoelectric session, a broaden feature is shown in Fig. 6d, which seems like the feature

in Fig. 2b. However, I can't see any clear "hint" in Fig. 5b. This is very important, since the authors also believe the "possible short-range magnetic interaction" here is coupled with the magnetoelectric effect. A high-resolution NPD experiment at low temperatures will solve all the magnetic structure related problems, and present a clear picture in this compound.

Reply:

We appreciate the feedback provided. First, we would like to acknowledge that the term 'short-range interaction' used in our previous manuscript is not an accurate expression of our point. The picture we tend to describe is a local canting that gives rise to local in-plane magnetic moments, and the local moments do not necessarily have short-range correlations. Therefore, we have removed the claim of possible short-range correlation in the revised manuscript.

We agree with the reviewer that a temperature dependent neutron powder diffraction is a worthy experiment to do. As shown in the new Fig. S4, the FWHM of the nuclear plus magnetic peak (0 2 1) shows a slight broadening tendency below T_N (Fig. S4g), while another nuclear plus magnetic peak (1 1 3) does not show systematic change of FWHM from 8 K to 40 K (Fig. S4f). We admit that this dataset cannot conclusively confirm the local canting picture, and future studies are necessary to unambiguously describe the local magnetization, but we would like to emphasize that the highlight of this work is the experimental discovery of large off-diagonal magnetoelectricity, while the ordered moments seem Ising collinear in neutron experiments and do not support an out-of-plane toroidal moment, which raises the reasonable expectation of the existence of local canted moments, and it is supported by the symmetry analysis considering the mirror symmetry breaking of Co^{2+} octahedrons.

Relevant discussion is added to the revised manuscript:

To probe the possible local symmetry breaking below T_N , NPD patterns were collected at intervals of 1 K during a ramping from 8 K to 40 K, and the obtained temperature contour plots of (1 1 3), (0 2 1), and (1 0 1) peaks are displayed in Fig. S4c to S4e. For the nuclear plus magnetic peak (0 2 1), the temperature-dependent full width at half maximum (FWHM) obtained from the Gaussian fitting of the peak profiles unveils a slight broadening tendency below T_N (Fig. S4g), while another nuclear plus magnetic peak (1 1 3) does not show systematic change of FWHM from 8 K to 40 K (Fig. S4f). So far, the data in the current work cannot be taken as conclusive evidence of local canting, which cannot be easily detected because only small magnetic components might be involved at the background level, and the major magnetic moment is ordered. In future studies, local-sensitive neutron technique, e.g., magnetic pair distribution function (*mPDF*) analysis could describe the local magnetic symmetry of $\text{CoTe}_6\text{O}_{13}$ in a more decisive manner [49]. (line 362 - 373)

Figure S4. (a) Neutron powder diffraction patterns taken on $\text{CoTe}_6\text{O}_{13}$ at 8 K, 14 K, and 40 K. (b) Magnetic peaks at 8 K and 40 K obtained by subtracting the 40 K pattern from the 8 K and 14 K patterns, respectively. (c)-(e) Temperature contour plots from a ramping experiment of (1 1 3), (0 2 1), and (1 0 1) peaks, respectively. (f)-(g) FWHM of (113) and (021) peaks as a function of temperature. The slicing interval of temperature is 3 K.

Some minor points:

- Define “ferro-rotation”.
- Line 157, missing “+” in Co^{2+} .
- Please use italics for the symbols of physical quantities.

Reply:

We greatly appreciate your thorough review. The minor issues you highlighted have been addressed in the revised version of the manuscript. Thank you for your attention to detail!

The overall science in this manuscript is fantastic. However, I believe the magnetic structure and low-temperature crystal structure are very important to the claims in this work. If the authors present more convinced experiment results in a revised version, I am happy to suggest its publication in *Nature Communications*.

Reply:

Thanks for the positive comments on our work! We appreciate your valuable suggestions. We hope the new experimental data and analysis added to the revised manuscript can address your concerns.

All changes in the manuscript are highlighted in yellow.

Reviewer #2 (Remarks to the Author):

I have reviewed this manuscript.

Xu et al. used a comprehensive set of experimental techniques to characterize poly- and single-crystalline samples of CoTe₆O₁₃. They described their observations in detail, complemented by some symmetry analysis.

The highlight is the observation of a large off-diagonal magnetoelectric effect in a long-ranged collinear spin structure, which is argued to be related to magnetic toroidicity.

Studies of this type are highly appreciated, as they assess an exotic phenomenon in physics – off-diagonal magnetoelectricity – via different angles, let alone the great potential held by multiferroics in device applications. The manuscript overall is well written and structured.

However, I have also identified items that need to be taken care of before I can recommend for publication. My overall judgements on these items are:

- They need to deliver more clear information.
- They need to be discussed in a more rigorous manner.

As a result, my recommendation is that the authors need to carry out a major revision to the current manuscript to address my concerns, and to meet the publication criteria of Nature Communications.

Reply:

Sincerely thanks for reviewing our work and giving valuable revision suggestions! We hope the new experimental data and analysis added to the revised manuscript can address your concerns.

I append my specific comments below, hoping that the authors can consider all of them when improving their manuscript.

1. In the abstract, the authors claim that 'Our results establish a symmetry criterion for a new mechanism of magnetic toroidicity, induced by collinear spins parallel to the toroidal moment'. In the second paragraph of the main manuscript, they also state that the origin of magnetic toroidicity in collinear spin systems is not straightforward. Based on these, it is natural for me to expect that they can shed new light on this puzzle in the current work. Unfortunately, the relevant discussions on symmetry are not very clear. For example, is canting, which is not necessarily related to the collinear structure, critical for generating toroidicity? I will comment on canting in more detail below.

Reply:

Thanks for the comment. Here is our opinion. For a collinear spin ground state, canting is not necessary to generate a toroidicity perpendicular to the spins, since a dimer distortion of the magnetic lattice can do it. However, canting seems necessary to generate a toroidicity parallel to the spins, because otherwise there will be no in-plane magnetic moments that can host that toroidicity.

Actually, the experimental observation on CoTe₆O₁₃ motivates us to propose this picture. First,

the single-crystal and powder neutron diffraction consistently give a $k = (0,0,0)$ and $R\bar{3}'$ magnetic structure, with collinear AFM spins parallel/antiparallel to c . However, the $\bar{3}'$ point group should

have magnetoelectric response $\sigma = \begin{bmatrix} \sigma_{11} & \sigma_{12} & 0 \\ -\sigma_{12} & \sigma_{11} & 0 \\ 0 & 0 & \sigma_{33} \end{bmatrix}$, where the off-diagonal term σ_{12} comes from

a toroidal moment along c , and experimentally we do observe a large off-diagonal magnetoelectricity. Then, there arises a question: If spins are strictly collinear along c , then the magnetic moment perpendicular to c is zero everywhere, and how can a toroidal moment along c exist?

To resolve this contradiction, a local canting picture seems the only possibility, which is also supported by the symmetry analysis, i.e., the missing mirror symmetries in the distorted Co^{2+} octahedron allows the spin canting away from the c axis. This canting happens within one octahedron; thus, it is local, and the resultant in-plane magnetic moments do not form a long-range order, thus this canting does not change the global symmetry $k = (0,0,0)$ and $R\bar{3}'$, and does not produce new diffraction spots in neutron diffraction. In other words, though the ordered moments look like out-of-plane collinear in neutron diffraction, in-plane moments are hiding there, forming a disordered steady state that produces a toroidal moment along c .

We admit that ‘Our results establish a symmetry criterion for a new mechanism of magnetic toroidicity, induced by collinear spins parallel to the toroidal moment’ is a bit overstating, and we have removed this sentence in the revised manuscript. Instead, we add “Based on the reported magnetic symmetries, the similar physics picture of toroidal moment parallel to collinear spins is also expected to exist in compounds such as MnTiO_3 ($\bar{3}'$) [54], GaFeO_3 ($m'm2'$) [55], $\text{TlFe}_{1.6}\text{Se}_2$ ($4/m'$) [56], $\text{K}_{0.8}\text{Fe}_{1.8}\text{Se}_2$ ($4/m'$) [57], CsCoF_4 ($\bar{4}'$) [58], KOsO_4 ($4'/m'$) [59], and KRuO_4 ($4'/m'$) [60], and the local magnetic symmetry breaking of these materials is worth testing in forthcoming experiments.” (line 404 - 412) to summarize other compounds that could similarly host toroidal moments parallel to the collinear spins, which may facilitate the future study of toroidal materials.

2. Fig. 1a displays a Le Bail fitting on the powder XRD data, i.e. profile matching. This is not a standard approach when fully solving a crystallographic structure. Can they present the Rietveld fitting instead? Or, if they choose to rely on the single crystal XRD refinement, they should plot those data accordingly, e.g. observed vs calculated intensities.

Reply:

Thanks for the good suggestion. We have added a new Fig. 1a and Table SIII showing the Rietveld refinement results:

Table SIII. The Rietveld refinement parameters of the lab powder x-ray diffraction on crushed $\text{CoTe}_6\text{O}_{13}$ crystals at room temperature (290 K). R -3 (No. 148), $\chi^2 = 4.14$. The lattice parameters $a = 10.1593(8)$ Å and $c = 18.9739(6)$ Å.

Atoms	x	y	z	B	Occ.	Site
Te1	0.82098(24)	0.57032(33)	0.73853(14)	0.256(36)	1	18f
Te2	0.60412(22)	0.50376(23)	0.42402(13)	0.000	1	18f
Co1	0.66667	0.33333	0.57727(65)	0.865(249)	1	6c
O1	0.66667	0.33333	0.74493(150)	0.000	1	6c
O2	0.74975(181)	0.54145(213)	0.64363(87)	0.000	1	18f
O3	0.61543(194)	0.47555(207)	0.51820(75)	0.137(476)	1	18f
O4	0.80003(224)	0.51626(174)	0.39883(84)	0.000	1	18f
O5	0.66347(200)	0.58736(197)	0.78976(99)	1.331(595)	1	18f

The $|F_{\text{obs}}|^2$ vs. $|F_{\text{cal}}|^2$ figure for SC-XRD refinement is also added to SI:

3. Fig. 2b, I am not convinced that one can directly/strongly argue short-range ordering based on the derivative of magnetic susceptibility. Is there any supporting literature, which used a similar approach to deliver this message, where the short-range ordering is also directly observed by, for example, neutron scattering?

Reply:

Thanks for raising this important point. One example that we came up with is the spinel Co_3O_4 . The coexisting of long-range order and short-range correlation below Neel temperature (~ 30 K) has been observed in neutron scattering (Zaharko et al, Physical Review B, 81 064416 (2010) and Golosova et al, Journal of Magnetism and Magnetic Materials, 508 166874 (2020)), and the reported $d\chi/dT$ of Co_3O_4 shows broad feature below Neel temperature (the inset of the figure attached below), similar to what we observed in $\text{CoTe}_6\text{O}_{13}$:

(Ikedo et al, Physical Review B, 75,054424 (2007))

We have added the new references and discussions to the revised manuscript. Thank you!

“A similar broad feature has been reported in the magnetic susceptibility of spinel Co_3O_4 [35], whose origin is believed to be a short-range incommensurate order evidenced by the diffuse scattering in neutron experiments [36, 37].” (line 164 - 166)

4. The authors also compared their magnetization data with those published. This was summarized in Fig. S4. In the caption, they state that they ‘guessed’ the magnetic field used in the previous published work. This should not be encouraged, as in peer reviewed publications, it is not appropriate to cite guessed measurement conditions. I suggest removing this comparison.

Reply:

Thanks for the good suggestion. We agree with the reviewer to remove the comparison in the revised manuscript.

5. Heat capacity measurements are used to support the effective spin at low temperatures and broad feature that might point to magnetic short-range ordering. This goes beyond simply determining the transition temperature. In practice, analysis of this kind requires a careful calibration. Subtraction of nonmagnetic contributions in the heat capacity data is critical for correctly reproducing the magnetic contributions in absolute unit, as well as the shape of its temperature dependence. In my opinion, these data provide the only hint in their work supporting that there is some short-range ordering below ~ 20 K at low field, which is buried in the noise of their neutron measurements. I would suggest the authors to carry out a separate nonmagnetic background measurement, e.g. on $\text{NMTe}_6\text{O}_{13}$, where NM is a nonmagnetic element. This is important because their discussion later is largely relying on the assumption that there is a local canting. Moreover, this also can further validate if the low temperature effective spin is indeed $3/2$. As the authors stated, in many other cases, Co ions in similar crystal field environment has an effective spin of $1/2$ at low temperatures.

Reply:

Thanks a lot for raising this important point. We have synthesized an isostructural non-magnetic compound $\text{MgTe}_6\text{O}_{13}$ and measured its heat capacity. Then, the magnetic contribution in the $\text{CoTe}_6\text{O}_{13}$ heat capacity is obtained by subtracting the $\text{MgTe}_6\text{O}_{13}$ heat capacity. Indeed, the obtained magnetic entropy of Co^{2+} is close to $R\ln 2$, i.e., a $S = 1/2$. It means that our previous phonon fitting curve is not accurate in the low-temperature regime, and the magnetic heat capacity was

overestimated. Sorry for this mistake. We have generated new Fig. 4 and modified relevant discussion:

Figure 4. (a) The heat capacity (C_p) of $\text{CoTe}_6\text{O}_{13}$ from 1.8 K to 40 K measured at 0 T, 3 T, 6 T, and 9 T field. The gray open squares display the heat capacity of the non-magnetic reference compound $\text{MgTe}_6\text{O}_{13}$. (b) The magnetic heat capacity over temperature (C_{mag}/T) obtained from subtracting the non-magnetic contribution. (c) The magnetic entropy versus temperature at 0 T obtained by integrating C_{mag}/T .

“The magnetic contribution (C_{mag}) to the heat capacity is obtained by subtracting the heat capacity of $\text{MgTe}_6\text{O}_{13}$ from the total heat capacity of $\text{CoTe}_6\text{O}_{13}$.” (line 230 - 231)

“It saturates above 30 K and reaches 5.75 J/mole- Co^{2+}/K . This value is close to the total magnetic entropy for a $S = 1/2$, $R\ln(2)$.” (line 235 - 236)

The experimental details of $\text{MgTe}_6\text{O}_{13}$ are also added:

“The isostructural non-magnetic compound $\text{MgTe}_6\text{O}_{13}$ was synthesized by a similar method but using a higher sintering temperature 630°C, with MgO (Alfa Aesar, 99.95%, 900°C overnight baked) and TeO_2 powders in a molar ratio 1:6 as the starting materials. The XRD pattern and Rietveld refinement result of $\text{MgTe}_6\text{O}_{13}$ are shown in Fig. S2c.” (line 83 - 87)

(New Fig. S2c)

6. Significant amount of information is missing in the presentation of their neutron diffraction data analysis.

a. For the audience, it is not straightforward to get the symmetry elements out by simply quoting the BNS symbols. Since the symmetry is important for understanding the magnetoelectric effect, please present a detailed symmetry analysis, e.g. irreducible representations and basis vectors.

Reply:

Thanks for pointing this out. We agree with the referee that simply quoting the BNS symbols are not enough for the audience. A detailed symmetry analysis based on irreducible representations are added into the revised manuscript:

“For crystallographic space group $R-3$ and propagation vector $k = (0, 0, 0)$, there are four possible magnetic irreducible representations, i.e., (i) mGM_1^+ , (ii) $mGM_2^+GM_3^+$, (iii) mGM_1^- , and (iv) $mGM_2^-GM_3^-$, resulting in the magnetic space groups $R-3$, $P-1$, $R-3'$, and $P-1'$, respectively. $R-3$ and $P-1$ belong to the ferromagnetic space group, which are not consistent with the observed antiferromagnetic ordering. Also, they do not break spatial inversion and do not allow linear magnetoelectricity, thus can be ruled out. Both $R-3'$ and $P-1'$ describe an antiferromagnetic ordering, while the spins in $R-3'$ are parallel/antiparallel to the three-fold axis, but $P-1'$ breaks the three-fold symmetry. Refining the magnetic structure using the $P-1'$ model obtained negligible in-plane magnetic moments, which suggests that the ordered moments are mostly along c . Overall, $R-3'$ (BNS #148.19) is the appropriate model to describe the antiferromagnetic ordering in $\text{CoTe}_6\text{O}_{13}$ ” (line 243 - 253).

We have also added Table SIV to summarize the Wyckoff positions of Co in those magnetic space groups, which are analyzed using the ‘k-subgroupsmag’ program in the Bilbao Crystallography Server.

Table SIV. Wyckoff positions of Co $(0, 0, z)$ and the allowed magnetic moment direction (m_x, m_y, m_z) in $R-3'$, $R-3$, $P-1'$, and $P-1$ magnetic space groups.

Magnetic space group	Wyckoff positions of Co
$R-3'$ (#148.19)	$(0,0,z \mid 0,0,m_z)$ $(0,0,-z \mid 0,0,-m_z)$ $(2/3,1/3,z+1/3 \mid 0,0,m_z)$ $(2/3,1/3,-z+1/3 \mid 0,0,-m_z)$ $(1/3,2/3,z+2/3 \mid 0,0,m_z)$ $(1/3,2/3,-z+2/3 \mid 0,0,-m_z)$
$R-3$ (#148.17)	$(0,0,z \mid 0,0,m_z)$ $(0,0,-z \mid 0,0,m_z)$ $(2/3,1/3,z+1/3 \mid 0,0,m_z)$ $(2/3,1/3,-z+1/3 \mid 0,0,m_z)$ $(1/3,2/3,z+2/3 \mid 0,0,m_z)$ $(1/3,2/3,-z+2/3 \mid 0,0,m_z)$
$P-1'$ (#2.6)	$(0,0,z \mid m_x,m_y,m_z)$ $(0,0,-z \mid -m_x,-m_y,-m_z)$ $(2/3,1/3,z+1/3 \mid m_x,m_y,m_z)$ $(2/3,1/3,-z+1/3 \mid -m_x,-m_y,-m_z)$ $(1/3,2/3,z+2/3 \mid m_x,m_y,m_z)$ $(1/3,2/3,-z+2/3 \mid -m_x,-m_y,-m_z)$
$P-1$ (#2.4)	$(0,0,z \mid m_x,m_y,m_z)$ $(0,0,-z \mid m_x,m_y,m_z)$ $(2/3,1/3,z+1/3 \mid m_x,m_y,m_z)$ $(2/3,1/3,-z+1/3 \mid m_x,m_y,m_z)$ $(1/3,2/3,z+2/3 \mid m_x,m_y,m_z)$ $(1/3,2/3,-z+2/3 \mid m_x,m_y,m_z)$

b. How many fitting parameters are there in the magnetic structure refinement? What are they? Is multi-domain configuration or spin canting already allowed by symmetry?

Reply:

In magnetic structure refinement, only four parameters are allowed: the isotropic Debye-Waller factors of Co, Te, and O, and the moment size of Co along c -axis. The scale factor, extinction factor and atom positions are fixed to the nuclear refinement result using the same set of peaks measured at 25 K.

The crystal is single domain as only one set of rhombohedral peaks are observed, and a global

spin canting is not allowed by the magnetic space groups. To allow a global spin canting, the magnetic symmetry needs to be reduced to $P-1'$. Using $P-1'$ symmetry in refinement yields a negligible in-plane magnetic moment of $0.2(2) \mu_B/\text{Co}^{2+}$ compared to the $4.4 \mu_B/\text{Co}^{2+}$ magnetic moment along c , which suggests that the ordered components of magnetic moments are along c without canting. Please refer to the reply to question #13 for more discussion on the possible local canting picture.

c. In Fig. 5a, how many points have non-negligible magnetic contributions? Perhaps, it is better to make a table, separately listing the nuclear and magnetic contributions at these locations, since $k = (0, 0, 0)$.

Reply:

In Fig 5a, 9 peaks have non-negligible magnetic contributions. Their nuclear contribution, magnetic contribution, and the experimentally observed intensities at 5 K are summarized in below table:

Table SV. Observed and calculated intensities of peaks having non-negligible magnetic contributions in single-crystal neutron diffraction.

(HKL)	F^2_{obs} at 5 K	F^2_{mag}	F^2_{nuc}	$F^2_{\text{mag+nuc}}$
(0 1 -1)	6.63 ± 0.42	7.06	0	7.06
(1 1 -3)	4.21 ± 0.12	4.25	0.44	4.69
(2 1 1)	6.15 ± 0.13	6.04	0.04	6.08
(-1 3 -1)	6.72 ± 0.33	6.04	0.04	6.08
(1 3 1)	4.11 ± 0.23	4.11	0.51	4.62
(2 0 5)	2.80 ± 0.08	2.29	0.13	2.44
(0 -2 5)	2.59 ± 0.07	2.29	0.13	2.44
(0 1 5)	1.02 ± 0.04	0.99	0.01	1.00
(-3 1 -1)	28.77 ± 0.58	5.78	25.71	31.49

7. The critical scaling analysis in Fig. 5b is incorrect, as this should only be applied to a narrow region near T_N , given that the data density there is sufficient.

Reply:

Thanks for pointing it out. We have revisited the critical scaling analysis and used only the

points above 15 K to fit the critical exponent. The correct exponent is $\beta = 0.28(5)$.

Figure 5(b) Temperature evolution of (0, 1, -1) magnetic peak intensity, the line is a power law fitting with the formula $I-I_0=A(1-T/T_N)^{2\beta}$. The point at 5 K is normalized from fitting the rocking curve scans of (0, 1, -1) peak at 5 K, 15 K and 25 K.

8. In Magnetolectricity, the authors state 'Moreover, the off-diagonal pyroelectric current in Fig. 6d shows a clear shoulder peak feature below TN. This may be related with a short-range magnetic ordering, which is also observed in the toroidal magnetolectric material LiNiPO4'. Since these measurements were performed in a sizable magnetic field, does the incomplete metamagnetic transition shown in Fig. 2c play a role in generating the short-range order?

Reply:

Thanks for the question. The off-diagonal magnetolectric measurement at 15 K (please refer to the reply for question #10) shows that the metamagnetic transition actually cause a decrease in the induced polarization, which implies that the metamagnetic transition suppresses the toroidal moment. We have added discussion to the revised manuscript:

As displayed in Fig. S7, the magnetolectric current loops collected at 2 K, 5 K, and 10 K are almost identical and show linear magnetolectricity within the -9 T to 9 T range, while anomalies in the magnetolectric response appear at around ± 7 T in the 15 K data, and the polarization in the high-field state is reduced. Those anomalies coincide with the metamagnetic transition shown in Fig. 2c, and this dataset implies that the metamagnetic transition suppresses the out-of-plane toroidal moment. (line 310 - 315)

9. In Magnetolectricity, the authors state 'Neutron diffraction (Fig. 5b) and magnetic susceptibility (Fig. 2b) data also show hints of that transition'. I cannot see such hints from the neutron diffraction

data presented. They will need to explain this statement. My comments on the susceptibility data can be found in Point 3 above.

Reply:

Sorry for the misinterpretation. To better reflect this point, we have performed a temperature-dependent neutron powder diffraction experiment, and the results are shown in the new Fig. S4. The FWHM of the nuclear plus magnetic peak (0 2 1) shows a slight broadening tendency below T_N (Fig. S4g), while another nuclear plus magnetic peak (1 1 3) does not show systematic change of FWHM from 8 K to 40 K (Fig. S4f). We admit that this dataset cannot conclusively confirm the local canting picture, and future studies are necessary to unambiguously describe the local magnetization, but we would like to emphasize that the highlight of this work is the experimental discovery of large off-diagonal magnetoelectricity, while the ordered moments seem Ising collinear in neutron experiments and do not support an out-of-plane toroidal moment, which raises the reasonable expectation of the existence of local canted moments, and it is supported by the symmetry analysis considering the local symmetry breaking of Co^{2+} octahedrons.

Relevant discussion is added to the revised manuscript:

To probe the possible local symmetry breaking below T_N , NPD patterns were collected at intervals of 1 K during a ramping from 8 K to 40 K, and the obtained temperature contour plots of (1 1 3), (0 2 1), and (1 0 1) peaks are displayed in Fig. S4c to S4e. For the nuclear plus magnetic peak (0 2 1), the temperature-dependent full width at half maximum (FWHM) obtained from the Gaussian fitting of the peak profiles unveils a slight broadening tendency below T_N (Fig. S4g), while another nuclear plus magnetic peak (1 1 3) does not show systematic change of FWHM from 8 K to 40 K (Fig. S4f). So far, the data in the current work cannot be taken as conclusive evidence of local canting, which cannot be easily detected because only small magnetic components might be involved at the background level, and the major magnetic moment is ordered. In future studies, local-sensitive neutron technique, e.g., magnetic pair distribution function (*mPDF*) analysis could describe the local magnetic symmetry of $\text{CoTe}_6\text{O}_{13}$ in a more decisive manner [49]. (line 362 - 373)

Figure S4. (a) Neutron powder diffraction patterns taken on $\text{CoTe}_6\text{O}_{13}$ at 8 K, 14 K, and 40 K. (b) Magnetic peaks at 8 K and 40 K obtained by subtracting the 40 K pattern from the 8 K and 14 K patterns, respectively. (c)-(e) Temperature contour plots from a ramping experiment of (1 1 3), (0 2 1), and (1 0 1) peaks, respectively. (f)-(g) FWHM of (113) peak and (021) peak as a function of temperature. The slicing interval is 3 K.

10. The magnetoelectric current loop measurements were done at 1.8 K (Fig. 7). If this effect is coupled to the magnetic structure, why aren't these measurements sensitive to the metamagnetic transition in Fig. 2c?

Reply:

Thanks for the good question. We have performed off-diagonal magnetoelectric measurements at higher temperatures, i.e., 5 K, 10 K, and 15 K. The metamagnetic transition (supposed to happen at around 7 T at 15 K) does have anomalies in the magnetoelectric current in the 15 K data. The metamagnetic transition at 1.8 K is not completed up to 9 T, and that's why we don't see it in the 1.8 K magnetoelectric data. The data is added to new Fig. S7.

Figure S7. (a) Off-diagonal magnetoelectric current measured at different temperatures (2 K, 5 K, 10 K, and 15 K). (b) The polarization as a function of time at 15 K.

Relevant discussion is added into the revised manuscript:

“As displayed in Fig. S7, the magnetoelectric current loops collected at 2 K, 5 K, and 10 K are almost identical and show linear magnetoelectricity within the -9 T to 9 T range, while anomalies in the magnetoelectric response appear at around ± 7 T in the 15 K data, and the polarization in the high-field state is reduced. Those anomalies coincide with the metamagnetic transition shown in Fig. 2c, and this dataset implies that the metamagnetic transition suppresses the out-of-plane toroidal moment.” (line 310 - 315)

11. In the Discussion, the authors say ‘As shown in the left panel of Fig. 8, if we introduce mirror symmetries (mirror planes parallel to c axis) into the Co^{2+} triangular lattice, i.e., removing the octahedron distortions, and keeping the magnetic structure unchanged...’. This assumption does not sound solid to me. From symmetry point of view, when you change the nuclear symmetry, the magnetic symmetry may also change. What they described following this assumption certainly is one plausible scenario, but the limitations of this assumption need to be addressed.

Reply:

Thanks for the comment. The reviewer is correct that when you change the nuclear symmetry, the magnetic symmetry may also change. Actually, it is exactly what we want to express, i.e., the oxygen octahedron having mirror symmetries or not, will determine the magnetic symmetry to be $-3'm'$ or $-3'$, respectively. ‘Keeping the magnetic structure unchanged’ seems not an accurate expression. We have replaced it by “keeping the Co^{2+} sublattice and spins unchanged” (line 328 - 329). Sorry for the confusion.

We have also addressed the limitation of our symmetry analysis in the modified manuscript:

“Note that, though the symmetry analysis in the present work suggests the existence of toroidal moments by locally canted collinear spins, it cannot give quantitative information on the magnitude of the induced toroidal moment. Therefore, a rigorous theoretical calculation would be helpful for a deeper understanding of the observed large off-diagonal magnetoelectricity.” (line 408 - 412)

12. ‘Therefore, collinear spins along c in the $-3'$ point group show symmetry operation similarity to

a toroidal moment along c formed by in-plane head-to-tail spin components.’ I cannot fully catch the point of this statement. If the authors believe that there is a connection between the two cases. A rigorous discussion is required.

Reply:

Thanks for the comment. The point we want to express is that the two spin textures in Fig. 8c have the same symmetry, i.e., they stay invariant under a three-fold rotation along c , or under a combination of spatial inversion and time reversal, and no other symmetry elements exist for both spin textures. Note that, if the octahedrons are not distorted, then the toroidal spin texture would have additional mirror symmetries than the collinear spin texture. That is why we believe the ferro-rotation type octahedron distortion is the key to enable canting. We have rephrased the sentences to better reflect our point: “Both the collinear spin texture (Fig. 8c left panel) and the toroidal spin texture (Fig. 8c right panel) only have three-fold rotational symmetry along c and a combination of spatial inversion and time reversal symmetry, without any additional symmetry elements such as mirror planes. Therefore, those two textures are equivalent in terms of symmetry operations, which suggests that the canting for an out-of-plane toroidal moment is allowed in $-3'$.” (line 344 - 349)

Thanks!

13. In the last part, the authors discussed short-ranged canting, and linked the system to other compounds reported in the literature. However, in my opinion, no clear evidence supporting canting is seen from their data. The first question one could ask is – is canting allowed by $k = (0, 0, 0)$? If not, a second propagation vector is required. Then the magnetic moment refinement presented above is not as accurate. Moreover, can short-ranged canting lead to such large off-diagonal magnetoelectricity? The effect reported in Ref. 42 is smaller.

Reply:

Thanks for the comments. The answer to the question ‘is canting allowed by $k = (0, 0, 0)$ ’ depends on the magnetic lattice. For example, as we briefly discussed in the manuscript, the reported magnetic structure of Yb_3Pt_4 is $k = (0, 0, 0)$ and $R-3'$ determined by neutron diffraction. Displayed in the figures below, the spins in Yb_3Pt_4 show in-plane moments from canting, and the in-plane moments form out-of-plane toroidal moments. So, in this case, $k = (0, 0, 0)$ definitely allows canting, and it can be clearly seen in neutron diffraction.

The situation in $\text{CoTe}_6\text{O}_{13}$ is a little bit different. The Co site locates at the center of each Yb_3 triangle. Thinking of merging those three Yb together into one higher symmetry site, that site should not contain a net in-plane moment. However, the toroidal feature should not disappear, and the spins should still have the freedom to rotate away from c . Therefore, in $\text{CoTe}_6\text{O}_{13}$ case, $k = (0, 0, 0)$ does not allow a global canting, but canting becomes an allowed local mode.

We did think about the possibility of having an additional k vector that accounts for an ordered canting and out-of-plane toroidal moment, for example, an in-plane $\sqrt{3} \times \sqrt{3}$ superlattice with $k=(1/3,1/3,0)$. However, no superlattice peak was observed in neutron diffraction. Therefore, we believe the local canting picture is the most likely one.

Since the ferro-rotation type structural distortion seems the key to produce spin canting, it is natural to expect that a larger structural distortion magnitude could generate a larger magnetic toroidicity. As shown in Fig. S1, $\text{CoTe}_6\text{O}_{13}$ has a much larger structural distortion magnitude than the ilmenite structure, which could explain the observed larger off-diagonal magnetoelectricity of $\text{CoTe}_6\text{O}_{13}$ than the reported off-diagonal magnetoelectricity of MnTiO_3 and $(\text{Ni,Mn})\text{TiO}_3$. In fact, off-diagonal magnetoelectricity was also reported in MnTiO_3 having the same $R\bar{3}'$ space group and $k = (0, 0, 0)$ (PRB, 90, 144429 (2014)), but the coefficient is ~ 100 times smaller than observed in $\text{CoTe}_6\text{O}_{13}$. Despite this, we are aware of the limitation of this qualitative thought, as we stated in the revised manuscript: “Note that, though the symmetry analysis in the present work suggests the existence of toroidal moments by locally canted collinear spins, it cannot give quantitative information on the magnitude of the induced toroidal moment. Therefore, a rigorous theoretical calculation would be helpful for a deeper understanding of the observed large off-diagonal magnetoelectricity.” (line 408 - 412)

14. Side comment: ac susceptibility measurements could be helpful to support their claim of short-range magnetic ordering.

Reply:

Thanks for the suggestion. We have performed the AC susceptibility measurement. Neither the real parts nor the imaginary parts of AC susceptibilities show detectable frequency dependence. Generally, AC susceptibility is more sensitive to probe spin freezing, like what happens in spin glasses. Here, though we believe that there exist local canted moments and they form a disordered steady state, but they may not have freezing feature in dynamics. That may be the reason why the AC susceptibility does not show frequency dependence.

Reviewers' Comments:

Reviewer #1:

Remarks to the Author:

The authors conducted more neutron scattering experiments to discuss the magnetic structure in this compound. I believe the revised manuscript can be published in Nature Communications.

Reviewer #2:

Remarks to the Author:

Referee report

I have reviewed this revised manuscript and response letter. I also read the report from Reviewer #1.

The authors have made much effort to address our concerns, including performing first-principle calculations. Many of the previous concerns have been addressed. However, in my opinion, the following three major concerns still exist. I cannot recommend publication in the current form.

1. Magnetic space group. In their initial submission, they made the following claim:

"Symmetry analysis yields two possible magnetic space groups: R -3 (BNS #148.19) and R -3' (BNS #148.17)".

In this revision, they have added two more space groups, which were not mentioned earlier. Their new statement reads:

"For crystallographic space group R-3 and propagation vector $k = (0, 0, 0)$, there are four possible magnetic irreducible representations, i.e., (i) mGM1+, (ii) mGM2+GM3+, (iii) mGM1-, and (iv) mGM2-GM3-, resulting in the magnetic space groups R-3, P-1, R-3', and P-1', respectively."

Then they proceed to describe the P-1' refinement, which allows in-plane canting and appears to be a plausible solution. From these descriptions, it is not convincing to me why this solution can be ruled out. For example, does it produce a poorer refinement quality? The authors need to make a concrete justification on their choice of R-3'.

2. Critical scaling. The choice of temperature window for scaling is arbitrary. For example, if an even narrower window is used, would it impact on the determination of beta? In practice, a log10-log10 plot between intensity and $(T_N - T)/T_N$ is required for a proper scaling analysis.

3. Spin canting. The authors argue that the in-plane canting in this system must be incoherent, or local. But they have observed a large off-diagonal magnetoelectric response from a bulk sensitive probe. Intuitively, I would expect that this kind of responses shall result from a coherent ferroelectric-type order parameter. It is not clear how an incoherent feature can lead to a strong coherent response.

End of report.

All changes in the manuscript are highlighted in yellow.

Reviewer #1

The authors conducted more neutron scattering experiments to discuss the magnetic structure in this compound. I believe the revised manuscript can be published in Nature Communications.

Reply:

We sincerely appreciate Reviewer #1 for the constructive efforts to enhance the quality of our manuscript and the positive feedback.

All changes in the manuscript are highlighted in yellow.

Reviewer #2 (Remarks to the Author):

I have reviewed this revised manuscript and response letter. I also read the report from Reviewer #1.

The authors have made much effort to address our concerns, including performing first-principle calculations. Many of the previous concerns have been addressed. However, in my opinion, the following three major concerns still exist. I cannot recommend publication in the current form.

Reply:

Thank you for your valuable comments! We have performed more experiments and analyses to address the remaining concerns.

1. Magnetic space group. In their initial submission, they made the following claim:

“Symmetry analysis yields two possible magnetic space groups: $R-3$ (BNS #148.19) and $R-3'$ (BNS #148.17)”.

In this revision, they have added two more space groups, which were not mentioned earlier. Their new statement reads:

“For crystallographic space group $R-3$ and propagation vector $k = (0, 0, 0)$, there are four possible magnetic irreducible representations, i.e., (i) $mGM1+$, (ii) $mGM2+GM3+$, (iii) $mGM1-$, and (iv) $mGM2-GM3-$, resulting in the magnetic space groups $R-3$, $P-1$, $R-3'$, and $P-1'$, respectively.”

Then they proceed to describe the $P-1'$ refinement, which allows in-plane canting and appears to be a plausible solution. From these descriptions, it is not convincing to me why this solution can be ruled out. For example, does it produce a poorer refinement quality? The authors need to make a concrete justification on their choice of $R-3'$.

Reply:

Thanks for the good question. We have carefully examined the possibility of the $P-1'$ model. In principle, the refinement using $P-1'$ should never give a poorer quality than using $R-3'$ because $P-1'$ is a lower-symmetry subgroup of $R-3'$. In other words, $R-3'$ can be redeemed as a special case of $P-1'$ when the global spin canting angle is approaching zero. Nevertheless, we believe that the ordered magnetic moments should be described by $R-3'$ for two reasons:

1. When we refine the single-crystal neutron diffraction data using $P-1'$, the obtained in-plane magnetic moment is $0.2(2) \mu_B$, which is negligible compared to the out-of-plane moment $4.4(1) \mu_B$. This result suggests that there is no need to pursue a further reduction in symmetry of $R-3'$.

2. The global spin canting in $P-1'$ breaks the rotational symmetry along c , thus it hosts in-plane toroidal moments in addition to the out-of-plane toroidal moment. Based on Neumann's principle and the MAGNDATA database, the linear magnetoelectric tensor of $P-1'$ has all nine non-zero

elements: $\sigma = \begin{bmatrix} \sigma_{11} & \sigma_{12} & \sigma_{13} \\ \sigma_{21} & \sigma_{22} & \sigma_{23} \\ \sigma_{31} & \sigma_{32} & \sigma_{33} \end{bmatrix}$, whereas the linear magnetoelectric tensor of $R-3'$

$\sigma = \begin{bmatrix} \sigma_{11} & \sigma_{12} & 0 \\ -\sigma_{12} & \sigma_{11} & 0 \\ 0 & 0 & \sigma_{33} \end{bmatrix}$ has $\sigma_{13} = \sigma_{23} = \sigma_{31} = \sigma_{32} = 0$. Experimentally, we have detected a large off-

diagonal σ_{12} linear magnetoelectric response on a single crystal plate with normal direction

perpendicular to c (Fig. S6a - S6d). Considering that having σ_{12} is not enough to distinguish $P-1'$ and $R-3'$, in the revised manuscript, we have performed the σ_{13} magnetoelectric measurement on a single crystal plate with the normal direction parallel to c (new Fig. S6e). As shown in new Fig. S6f, no hint of pyroelectric anomaly at T_N has been detected in this σ_{13} measurement setup, suggesting the absence of σ_{13} element in the magnetoelectric tensor, and the only off-diagonal linear magnetoelectric response in $\text{CoTe}_6\text{O}_{13}$ is σ_{12} . This observation confirms that the ordered moments in $\text{CoTe}_6\text{O}_{13}$ are parallel to c , and $R-3'$ is the appropriate model.

We have addressed it in the revised manuscript: “Refining the magnetic structure using the $P-1'$ model obtained negligible in-plane magnetic moments ($0.2(2) \mu_B$), which suggests that the ordered moments are mostly along c . The absence of σ_{13} in the experimental magnetoelectric tensor (Fig. S6) also provides evidence against the $P-1'$ model.” (lines 251 - 254)

Figure S6. The magnetoelectric tensor elements σ_{12} (a - d) and σ_{13} (e - f) measurements. (a) A photo of the single crystal plate with electrodes used for σ_{12} measurement. Directions of measured polarization, applied magnetic field, and c axis are marked by yellow, blue, and red arrows, respectively. (b) Magnetoelectric current at 1.8 K measured with magnetic field perpendicular to the current, and both magnetic field and current are perpendicular to the c axis. (c) The sweeping magnetic fields and electric polarizations obtained from (b) as a function of time. (d) The pyroelectric current (lower panel) and obtained polarization (upper panel) as a function of temperature in σ_{12} setup. (e) A photo of the single crystal plate with electrodes used for σ_{13} measurement. (f) The pyroelectric current as a function of temperature in the setup for σ_{13} measurement, which does not show detectable anomaly. Note that the current signals measured on the single crystals are noisier than those on the polycrystalline sample (Fig. 7 in the main text) because the area of the single crystal sample is limited by its smaller size.

2. Critical scaling. The choice of temperature window for scaling is arbitrary. For example, if an even narrower window is used, would it impact on the determination of beta? In practice, a log10-log10 plot between intensity and $(T_N - T)/T_N$ is required for a proper scaling analysis.

Reply:

Thank you for bringing up this issue. We agree with Reviewer #2 that the critical analysis should ideally be done within the narrow vicinity of the transition. However, our current fitting includes seven data points below the transition, and further reducing the fitting range would result in a less reliable fitting results due to limited data points. We have addressed it in the revised manuscript: “Notably, due to limited data points collected near the T_N , we believe that our fitting only serves as a guidance for the measured data rather than yielding a definitive critical exponent β .” (lines 270 - 272) Thanks for your kind understanding.

3. Spin canting. The authors argue that the in-plane canting in this system must be incoherent, or local. But they have observed a large off-diagonal magnetoelectric response from a bulk sensitive probe. Intuitively, I would expect that this kind of responses shall result from a coherent ferroelectric-type order parameter. It is not clear how an incoherent feature can lead to a strong coherent response.

Reply:

Thanks for the comment. Here are our thoughts. The magnetoelectricity (both diagonal and off-diagonal) in $\text{CoTe}_6\text{O}_{13}$ is a symmetry-protected linear magnetoelectricity, since the experimentally observed polarization shows a good linear relationship to the applied magnetic field in the whole -9 T to 9 T range. Also, the observed linear magnetoelectric tensor is consistent with the one expected for $R\bar{3}'$. Therefore, we believe that the magnetically ordered state in $\text{CoTe}_6\text{O}_{13}$ does host magnetic ferro-toroidicity as a ferroic order, which generates the off-diagonal linear magnetoelectricity. However, this ferro-toroidicity may be a secondary order parameter induced by the combination of the structural ferro-rotation order and the A-type collinear antiferromagnetic order detected in neutron diffraction. Using an example, if we apply a ferro-rotation type structural distortion (e.g., using a shear strain) to a linear diagonal magnetoelectric material such as Cr_2O_3 , a ferro-toroidicity should be expected to appear, since the linear diagonal magnetoelectricity maps the electric dipole vortex into a spin vortex! The only difference is that the ferro-rotation structural distortion is intrinsic in $\text{CoTe}_6\text{O}_{13}$. The spin local canting hypothesis serves as a means to visualize the microscopic picture and reflect the secondary order nature of the toroidicity -- it does not imply that the observed toroidicity arises from randomness. Thanks for reading!

End of report.

Reviewers' Comments:

Reviewer #2:

Remarks to the Author:

I have read the response letter and revised manuscript.

The authors have performed additional measurements on σ_{13} . The results obtained support their choice of R-3 in symmetry. They have also softened their statement on the critical scaling. Moreover, they made a phenomenological discussion on the appearance of linear ME effect in CoTe₆O₁₃, which I find self-consistent.

I greatly appreciate the authors' effort to address my concerns in full. I recommend publishing their work.

REVIEWERS' COMMENTS

Reviewer #2 (Remarks to the Author):

I have read the response letter and revised manuscript.

The authors have performed additional measurements on σ_{13} . The results obtained support their choice of $R-3$ in symmetry. They have also softened their statement on the critical scaling. Moreover, they made a phenomenological discussion on the appearance of linear ME effect in $\text{CoTe}_6\text{O}_{13}$, which I find self-consistent.

I greatly appreciate the authors' effort to address my concerns in full. I recommend publishing their work.

Reply: Thank you for taking the time to review our response letter and the revised manuscript. We're pleased to hear that you've found the additional measurements on σ_{13} supportive of our choice of $R-3$ symmetry and that the adjustments made to our statements on critical scaling align with your expectations. Your positive feedback on the phenomenological discussion of the linear ME effect in $\text{CoTe}_6\text{O}_{13}$ is encouraging. In future studies, we will further investigate this intriguing system using more advanced techniques to gain a deeper understanding of the microscopic physics picture.

Your recommendation to publish our work means a lot to us, and we are grateful for your support and guidance throughout this process. Thank you sincerely.